# FRONT-LOADING REASONING: THE SYNERGY BETWEEN PRETRAINING AND POST-TRAINING DATA

**Syeda Nahida Akter**[2],[*], **Shrimai Prabhumoye**[1],[3], **Eric Nyberg**[2], **Mostofa Patwary**[1],
**Mohammad Shoeybi**[1], **Yejin Choi**[1],[4], **Bryan Catanzaro**[1]
NVIDIA[1], Carnegie Mellon University[2], Boston University[3], Stanford University[4]
`sakter@andrew.cmu.edu, sprabhumoye@nvidia.com`

## ABSTRACT

The prevailing paradigm for enhancing the reasoning abilities of Large Language Models (LLMs) revolves around post-training on high-quality, reasoning-intensive data. While emerging literature suggests that reasoning data is increasingly incorporated also during the mid-training stage—a practice that is relatively more proprietary and less openly characterized—the role of such data in pretraining remains unclear. In particular, due to the opaqueness of pretraining corpora in most frontier models, the effect of reasoning data introduced at different phases of pre- and/or post-training is relatively less reported in the scientific literature. This raises several important but unsettled questions: *Is adding reasoning data earlier during pre-training any better than introducing it during post-training, when the token counts are controlled? Could earlier inclusion risk overfitting and harm generalization, or instead establish durable foundations that later fine-tuning cannot recover?* To address these questions, we conduct the first systematic study of *how* reasoning data—varying in scale, diversity, and quality—affects LLM performance *when* introduced at different stages of training. Our findings reveal that *front-loading reasoning data into pretraining is critical (19% average gain)*, establishing foundational capabilities that cannot be fully replicated by later-stage SFT, even with more data. We uncover an asymmetric principle for optimal data allocation: *pretraining benefits most from broad diversity in reasoning patterns (11% average gain), while SFT is more sensitive to data quality (15% average gain with high quality data)*. Furthermore, we show that *high-quality pretraining data has latent effects, activated only after SFT*, and that *naively scaling SFT data can be detrimental*, washing away the benefits of early reasoning injection. Collectively, our results challenge the conventional separation of language modeling and reasoning, providing a principled guide for strategically allocating data across the entire training pipeline to build more capable models.

## 1 INTRODUCTION

The reasoning abilities of Large Language Models (LLMs) have advanced considerably, with post-training on reasoning data driving significant breakthroughs in reasoning tasks, such as math competitions (Hendrycks et al., 2021b), PhD-level scientific QA (Rein et al., 2024; Phan et al., 2025), and software engineering (Jimenez et al., 2024). This progress has been largely driven by mid- or post-training LLMs on high-quality, reasoning-intensive datasets—often featuring long chain-of-thought (CoT) examples (Guha et al., 2025; Moshkov et al., 2025; Zhou et al., 2025; Gandhi et al., 2025; Wang et al., 2025). While this approach has proven effective, it treats reasoning as a specialized skill to be layered onto a generalist base. In addition, the impact of incorporating reasoning data during pretraining—and the potential synergistic effects on subsequent post-training—remains a critical yet less explored frontier. This research gap persists due to the prohibitive computational cost of end-to-end pretraining experiments and the opacity surrounding proprietary training recipes, which has concentrated community efforts on the more accessible post-training phase.

---

[*]Work done during internship at NVIDIA

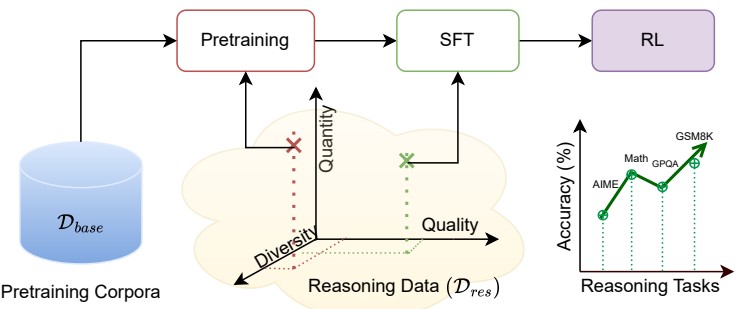

Figure 1: We systematically inject reasoning-style data ($\mathcal{D}_{\text{res}}$) at different phases of training—pretraining versus SFT—while varying its *diversity, quantity, and quality*. Our results show an asymmetric principle: diversity and scale matter most during pretraining, whereas quality dominates in SFT. This allocation strategy compounds through reinforcement learning (RL), yielding sustained gains across complex reasoning benchmarks.

The synergy between post-training phases has been widely explored (Liu et al., 2025; Chen et al., 2025b; Chu et al., 2025), yet conclusions vary with training data and scale, and their applicability to pretraining remains vague in the current literature. In this work, we investigate not just *which* reasoning data, but *when* to train with such reasoning data by studying the synergy between pretraining and post-training. Our central goal is to determine the ideal balance of such reasoning data across the two phases in order to maximize downstream accuracies after reinforcement learning. This motivates the following research questions:

- *Is a reasoning-rich pretraining essential, or can a model "catch up"?* We investigate whether a model pretrained without reasoning data can match the performance of its reasoning-aware counterparts by simply undergoing a more intensive SFT phase.

- *Does inclusion of reasoning data make the base LLM overfitted and less generalizable to sustain gains in subsequent training phases?* While recent literature highlights overspecialization of reasoning during post-training can be detrimental (Gupta et al., 2025; Luo et al., 2025b), investigations of this effect in pretraining remain limited.

- *Does data diversity in pretraining impact stability and specialization during SFT?* Specifically, does using the same reasoning data in both pretraining and SFT lead to robust skill mastery, or does a *narrow pretraining focus* risk catastrophic forgetting when the model is later fine-tuned?

- *Does the complexity and quality of reasoning data matter when incorporated during pretraining of the base model?* Current literature explores this mostly from SFT stage (Zhou et al., 2023; Guha et al., 2025), making it obscure whether difficulty or noisiness in the early phase of training directly impacts reasoning development or not.

This work provides a systematic analysis of the interplay between reasoning data and the distinct phases of LLM training. Our primary findings are summarized as:

- **Front-loading reasoning data into pretraining creates a durable, compounding advantage.** Injecting reasoning data during pretraining establishes a superior foundation that widens at every stage of post-training, culminating in a **+19%** lead on expert-level benchmarks. This refutes the *catch-up* and *overfitting* hypotheses, proving that SFT cannot compensate for a weak foundation and that pretraining choices dictate the final performance ceiling.

- **The optimal data strategy is asymmetric: prioritize diversity in pretraining and quality in SFT.** Our results reveal a clear, phase-dependent principle. Pretraining benefits most from **diversity and scale** (a **+11%** gain with diverse corpus), while SFT is dominated by **data quality** (a **+15%** gain with high-quality reasoning data). This provides an actionable heuristic for data allocation that is more nuanced than simplistic "more is better" approaches.

- **Naive scaling of SFT data is ineffective and harmful.** Blindly scaling SFT with mixed-quality data yields no average improvement and actively harmed mathematical reasoning by **-5%** on average, while a marginal (0.4%) addition of high-quality data consistently improved performance.

- **High-quality pretraining data can have a latent effect unlocked by SFT.** We found that high-quality data added to a diverse pretraining mix showed minimal immediate benefit but "unlocked" an additional **+4%** gain over model pretrained with diverse, mixed quality data after SFT—revealing a deeper synergy where pretraining can instill a latent potential in the model that is only activated during the alignment phase.

## 2 METHODOLOGY

Our methodology is designed to systematically determine the optimal strategy for allocating reasoning data between the pretraining and supervised fine-tuning stages of LLM development. We frame this as an optimization problem where the goal is to maximize the final model's downstream accuracies, $\mathcal{P}$. This is a function of the reasoning data introduced during pretraining, $\mathcal{D}_{\mathrm{res}}^{\mathrm{PT}}$, and the data used for supervised fine-tuning, $\mathcal{D}_{\mathrm{res}}^{\mathrm{SFT}}$. Our objective is to find the optimal data configurations, $(\mathcal{D}_{\mathrm{res}}^{\mathrm{PT}*}, \mathcal{D}_{\mathrm{res}}^{\mathrm{SFT}*})$, that solves the following:

$$(\mathcal{D}_{\mathrm{res}}^{\mathrm{PT}*}, \mathcal{D}_{\mathrm{res}}^{\mathrm{SFT}*}) = \arg \max_{\mathcal{D}_{\mathrm{res}}^{\mathrm{PT}}, \mathcal{D}_{\mathrm{res}}^{\mathrm{SFT}}} \mathcal{P}(\theta_{\mathrm{final}})$$

where $\theta_{\mathrm{final}}$ represents the parameters of the final model trained on data recipes defined by the choice of reasoning data at both stages.

Let $\mathcal{D}_{\mathrm{base}}$ denote the general pretraining corpus and we define a model $\mathcal{M}(\theta)$ with parameters $\theta$ trained in two stages:

$$\textbf{Pretraining:} \quad \theta_{\mathrm{PT}} = \arg \min_{\theta} \ \mathbb{E}_{(x,y) \sim \mathcal{D}_{\mathrm{base}} \cup \mathcal{D}_{\mathrm{res}}^{\mathrm{PT}}} \ \mathcal{L}_{\mathrm{LM}}(f_{\theta}(x), y),$$

$$\textbf{SFT:} \quad \theta_{\mathrm{SFT}} = \arg \min_{\theta} \ \mathbb{E}_{(x,y) \sim \mathcal{D}_{\mathrm{res}}^{\mathrm{SFT}}} \ \mathcal{L}_{\mathrm{SFT}}(f_{\theta}(x), y),$$

**Evaluation Objective.** The central research question can be expressed as analyzing the function:

$$\mathcal{P}(\mathcal{D}_{\mathrm{res}}^{\mathrm{PT}}, \mathcal{D}_{\mathrm{res}}^{\mathrm{SFT}}) = \mathbb{E}_{t \sim \mathcal{T}} \Big[ \mathrm{Acc}\big(f_{\theta_{\mathrm{SFT}}}(t)\big) \Big], \tag{1}$$

where $\mathcal{T}$ is a set of downstream reasoning tasks (math, science, code, general reasoning) and the expectation $\mathbb{E}_{t \sim \mathcal{T}}$ defines a single fine-tuned model that is evaluated across $\mathcal{T}$.

Our study can be summarized as optimizing the allocation of $\mathcal{D}_{\mathrm{res}}$ between pretraining and SFT:

$$\max_{\mathcal{D}_{\mathrm{res}}^{\mathrm{PT}}, \mathcal{D}_{\mathrm{res}}^{\mathrm{SFT}}} \mathcal{P}(\mathcal{D}_{\mathrm{res}}^{\mathrm{PT}}, \mathcal{D}_{\mathrm{res}}^{\mathrm{SFT}}) \quad \text{s.t.} \quad \mathcal{B} = |\mathcal{D}_{\mathrm{res}}^{\mathrm{PT}}| + |\mathcal{D}_{\mathrm{res}}^{\mathrm{SFT}}|, \tag{2}$$

where $\mathcal{B}$ is the total budget of reasoning data available. This captures the trade-off of early, scale/diversity vs late, quality/complexity: $\mathcal{D}_{\mathrm{res}}^{\mathrm{PT}} \longleftrightarrow \mathcal{D}_{\mathrm{res}}^{\mathrm{SFT}}$

### 2.1 MODEL ARCHITECTURE AND BASELINE

We select a hybrid transformer with a mixture of Mamba 2 (Dao & Gu, 2024), self-attention and FFN layers (NVIDIA, 2025a) with an 8B parameter for our base model, $\mathcal{M}$ and pretrain from scratch for 1 trillion tokens. This size strikes a balance between computational feasibility and the capacity to learn complex reasoning patterns.

### 2.2 DATA PIPELINE

Our experimental design relies on a careful distinction between two categories of data: (1) a large-scale, general-purpose pretraining corpus, and (2) a reasoning-focused, instruction-tuning (SFT-style) datasets of varying quality and scale. This separation allows us to precisely control the injection of reasoning data at different stages of training.

**General Pretraining Corpus ($\mathcal{D}_{\mathrm{base}}$).** For the base training corpus, we adopt the dataset introduced in NVIDIA (2025b), which contains 6.2T tokens drawn from high-quality Common Crawl, mathematics, and code sources. This corpus provides broad coverage of languages and technical domains, serving as the backbone of all pretraining experiments.

**Reasoning Datasets ($\mathcal{D}_{\mathrm{res}}$).** To investigate the impact of data quality, diversity, and complexity, we curate four distinct reasoning-focused datasets in the question-answer format:

- **Large-Scale, Diverse Data ($\mathcal{D}_{\mathrm{LDQ}}$).** To simulate a *"quantity-over-quality"* strategy, we employ the Nemotron-Pretraining-SFT-v1 dataset (NVIDIA, 2025b). This massive 268M samples of dataset offers extensive domain coverage, with a composition of approximately 56% math, 17% code, and 27% science and general-purpose reasoning. The dataset covers tasks ranging from simple Q&A to multi-turn dialogues, but with heterogeneous quality and reasoning depth, reflecting large-scale real-world availability.

- **Small-Scale, High-Quality Data ($\mathcal{D}_{\mathrm{SHQ}}$).** To capture the effect of long chain-of-thought traces from strong teacher models, we include the dataset of Guha et al. (2025), comprising 1.2M carefully curated examples (71% math, 21% code, 8% science). Compared to $\mathcal{D}_{\mathrm{LMQ}}$, this corpus is smaller, less diverse, but significantly higher quality, emphasizing detailed reasoning paths.

- **Large-Scale, Mixed-Quality Data ($\mathcal{D}_{\mathrm{LMQ}}$).** To balance diversity with quality, we construct a combined dataset that is a straightforward union of the two datasets above: $\mathcal{D}_{\mathrm{LMQ}} = \mathcal{D}_{\mathrm{LDQ}} + \mathcal{D}_{\mathrm{SHQ}}$, i.e., $\mathcal{D}_{\mathrm{LMQ}}$ contains 269.2M samples in total. This mix preserves large-scale coverage while injecting a fraction of curated, high-quality reasoning traces.

- **Answer-Length Filtered Data ($\mathcal{D}_{\mathrm{ALF}}$).** To investigate the feature of data quality, we create a subset (7.1M samples) of $\mathcal{D}_{\mathrm{LDQ}}$ by retaining examples where the answer length exceeds 4096 tokens, based on the principle that longer responses often correspond to more complex CoT reasoning. This dataset allows us to isolate the impact of reasoning complexity in different training phases.

## 2.3 Synergy between Pretraining and SFT

In this work, we aim to disentangle the contribution of reasoning data when incorporated at different points in the training pipeline. We structure the study into three stages: (i) large-scale **Pretraining**, where reasoning data may or may not be injected alongside the base corpus, (ii) **Supervised Finetuning (SFT)**, where pretrained models are further adapted on reasoning data of varying quality and diversity, and (iii) **Reinforcement Learning (RLVR)** to determine the sustainability of early reasoning gain in the final model.

**Phase 1: Pretraining.** Prior work has primarily explored reasoning supervision either on top of fully pretrained LLMs (Wang et al., 2025) or by introducing small amounts of long chain-of-thought (CoT) data into intermediate checkpoints (AI et al., 2025). These approaches leave open two questions: how to inject reasoning data at scale during end-to-end pretraining, and whether the benefits persist when combined with high-quality base corpora. To address these questions, we pretrain all models *from scratch* for 1T tokens using a mix of $\mathcal{D}_{\mathrm{base}}$ and different types of $\mathcal{D}_{\mathrm{res}}$. Across all models, we keep the token ratio between $\mathcal{D}_{\mathrm{base}}$ and $\mathcal{D}_{\mathrm{res}}$ fixed during pretraining. When a reasoning dataset is small, it is repeated so that the model still observes the same total volume of reasoning tokens. To correctly state the data distribution, we pretrain all models for 600B tokens using $\mathcal{D}_{\mathrm{base}}$ followed by 400B tokens on a mixture of 80% $\mathcal{D}_{\mathrm{base}}$ and 20% $\mathcal{D}_{\mathrm{res}}$. This results in a constant budget of 80B reasoning tokens across all experiments. $\mathcal{D}_{\mathrm{res}}$ can be any one source of data among the three reasoning datasets: $\mathcal{D}_{\mathrm{SHQ}}$, $\mathcal{D}_{\mathrm{LDQ}}$, $\mathcal{D}_{\mathrm{LMQ}}$. This token ratio has been maintained across all three pretraining runs with reasoning data.

Based on the reasoning data introduced, we train four distinct models:

- $\mathcal{M}_{\mathrm{base}}$: This model serves as our **baseline** and is pretrained without any reasoning data.
- $\mathcal{M}_{\mathrm{LDQ}}$: Pre-trained with large-scale, diverse $\mathcal{D}_{\mathrm{LDQ}}$ reasoning dataset along with $\mathcal{D}_{\mathrm{base}}$.
- $\mathcal{M}_{\mathrm{SHQ}}$: Pre-trained with $\mathcal{D}_{\mathrm{SHQ}}$ and $\mathcal{D}_{\mathrm{base}}$ allowing us to isolate the effect of data quality versus the quantity and diversity of $\mathcal{M}_{\mathrm{LDQ}}$.
- $\mathcal{M}_{\mathrm{LMQ}}$: Finally, this model is exposed to our combined reasoning $\mathcal{D}_{\mathrm{LMQ}}$ dataset.

In the subsequent analysis, we use $\mathcal{M}_{\mathrm{res}}$ to denote the aggregate performance of the models pretrained with reasoning data, representing the average score across $\mathcal{M}_{\mathrm{SHQ}}$, $\mathcal{M}_{\mathrm{LDQ}}$, and $\mathcal{M}_{\mathrm{LMQ}}$.

**Phase 2: Supervised Finetuning.** Following pretraining, each of the four model variants ($\mathcal{M}_{\mathrm{base}}$, $\mathcal{M}_{\mathrm{LDQ}}$, $\mathcal{M}_{\mathrm{SHQ}}$, $\mathcal{M}_{\mathrm{LMQ}}$) is adapted through supervised finetuning (SFT). This second phase is

crucial for understanding the synergies, redundancies, and trade-offs between the data introduced during pretraining versus the SFT stage. To this end, we design a controlled set of SFT experiments, where each pretrained model is finetuned on different reasoning corpora introduced in Section 2.2 to address the following rearch questions:

- **The "Catch-Up" Hypothesis:** Can intensive SFT on high-quality reasoning data allow the baseline model, $\mathcal{M}_{\text{base}}$, to match or exceed the accuracy of models that were exposed to reasoning data during pretraining? This directly tests the criticality of early data injection versus late-stage specialization.

- **Impact of Pretraining Data Scale and Diversity:** We investigate how the scale and diversity of reasoning data used during pretraining influence the final model's capacity to absorb high-quality instruction data. Specifically, we ask: *Does scaling up diverse reasoning data in pretraining provide lasting benefits even after all models are finetuned on the same high-quality SFT corpus?* By fine-tuning both the model pretrained on large, diverse data ($\mathcal{M}_{\text{LDQ}}$) and on smaller, less diverse data ($\mathcal{M}_{\text{SHQ}}$) on the same high-quality SFT set, we can determine whether a broad or a deep initial exposure to reasoning yields a better foundation for downstream specialization.

- **Impact of SFT Data Quality and Complexity:** By fine-tuning all four base models on datasets of varying quality ($\mathcal{D}_{\text{LDQ}}$ vs. $\mathcal{D}_{\text{SHQ}}$) and complexity ($\mathcal{D}_{\text{ALF}}$), we can measure the marginal utility of data quality at the SFT stage as a function of the model's initial pretraining condition.

This design enables us to address three critical dimensions: (1) the **synergy** between pretraining and SFT data, (2) the **gains** of increasing SFT data scale when reasoning was already introduced in pretraining, and (3) the **role of data complexity and diversity** in determining whether reasoning supervision should be injected early, late, or across both stages. Together with the pretraining experiments, these SFT studies form a fully crossed setup, providing the first systematic assessment of how reasoning-style SFT data interacts with pretraining to shape the reasoning abilities of LLMs.

**Phase 3: Reinforcement Learning.** To further observe the impact of reasoning centric pretraining and heavy supervised finetuning, we deploy RL using Group Relative Policy Optimization (GRPO) (Shao et al., 2024) with verifiable rewards on top of the base models. Here we use NEMOTRON-CROSSTHINK (Akter et al., 2025) which has shown to be effective for multi-domain reasoning.

## 3 EXPERIMENTAL SETUP

### 3.1 TRAINING

**Pretraining.** To prepare base models, we pretrain a 8B LLM on our pretraining data blend till 1T tokens using 512 H100 80GB SXM5 GPUs. During training, we use the AdamW optimizer (Loshchilov & Hutter, 2019) with $\beta_1 = 0.9$, $\beta_2 = 0.95$ and weight decay of 0.1. We use a 8-way tensor and pipeline parallelism to train the model. We set the maximum value of learning rate to $3e^{-4}$, minimum to $3e^{-6}$, and use a batch size of 6M tokens with a 8192 context length.

**Post-Training.** After pretraining, each 8B LLM is finetuned on 4.8M reasoning samples from $\mathcal{D}_{res}$. SFT uses AdamW with $(\beta_1, \beta_2) = (0.9, 0.95)$, weight decay 0.01, warmup ratio 0.05, learning rate $5 \times 10^{-6}$, batch size 512, and context length 32k. We then apply GRPO via the veRL framework[1] for one epoch on NEMOTRON-CROSSTHINK data with constant LR $1 \times 10^{-6}$, batch size 128, PPO mini-batch 128, and context length 8192. Each step samples 128 prompts with 8 rollouts (temperature= 1.0, top-$p$ = 1.0), and a KL penalty coefficient of 0.001.

### 3.2 EVALUATION METRICS

We report average accuracies of all tasks under each of the following categories.

**Base Model Evaluations.** We conduct a thorough benchmark assessment to evaluate the generalizability of the base models, using a series of datasets using LM Eval Harness (Gao et al., 2024).

---

[1]https://github.com/volcengine/verl

- **General Purpose Reasoning (GPR$_{\text{PT}}$ AVG).** We consider four standard commonsense and logical reasoning tasks in 0-shot: ARC challenge (Clark et al., 2018), HellaSwag (Zellers et al., 2019), WinoGrande (Sakaguchi et al., 2021), and reading comprehension task: RACE (Lai et al., 2017).

- **Math Reasoning (MATH$_{\text{PT}}$ AVG).** We evaluate the math reasoning ability with two benchmarks– they encompass math challenges from elementary to college level complexity demanding qualitative reasoning (8-shot GSM8K (Cobbe et al., 2021), 4-shot MATH-500 (Hendrycks et al., 2021b)).

- **Science Reasoning (SCIENCE$_{\text{PT}}$ AVG).** We evaluate on 5-shot MMLU (Hendrycks et al., 2021a) and MMLU-Pro (Wang et al., 2024) that spans multiple domains, from professional to academic, testing the model on specialized subjects.

- **Code Reasoning (CODE$_{\text{PT}}$ AVG).** For code tasks (HumanEval (Chen et al., 2021), MBPP (Odena et al., 2021)) we evaluate the EvalPlus variants along with the sanitization of generations (Liu et al., 2023), in a 0-shot setup. We estimate avg@32, pass@1 from 32 generations per prompt.

**SFT Model Evaluations.** To evaluate the reasoning ability of different SFT models, we focus on reasoning centric benchmarks unlike in base model evaluations, where mostly focus on the generalizability of the LLM. We conduct evaluations using NeMo-Skills[2].

- **Math Reasoning (MATH$_{\text{SFT}}$ AVG).** In addition to the GSM8K and MATH-500, we evaluate the models on two more complex math tasks—AIME24 and AIME25 (Veeraboina, 2023).

- **Science Reasoning (SCIENCE$_{\text{SFT}}$ AVG).** On top of MMLU and MMLU-Pro, we evaluate on graduate level QA task: GPQA-Diamond (Rein et al., 2024).

- **Code Reasoning (CODE$_{\text{SFT}}$ AVG).** We choose LiveCodeBench (Jain et al., 2025) to test complex code reasoning ability.

- **Instruction Following (INS$_{\text{SFT}}$ AVG).** For broader evaluation on diverse capabilities, we use IFEval (Zeng et al., 2024).

We report Pass@1 average of 16 runs for AIME-2024, AIME-2025 and average of 4 runs for MATH-500, GSM8K, MMLU, MMLU-Pro, GPQA-Diamond, LiveCodeBench and IFEval.

**RL Model Evaluations.** In this phase, we evaluate the models on complex reasoning tasks such as AIME24,25, MATH-500, GSM8K, MMLU, MMLU-Pro, GPQA-Diamond, LiveCodeBench following the evaluation metric in SFT phase.

# 4 EXPERIMENTS AND RESULTS

**Immediate Foundational Gains from Reasoning Data in Pretraining.** Table 1 shows the average accuracies of our four model variants immediately after the 1T token pretraining phase (see Table 9 for individual benchmarks). The results provide clear evidence that integrating reasoning-style corpora from the start builds a significantly more capable foundation. Every model exposed to

| Model | Average | MATH$_{\text{PT}}$ AVG | SCIENCE$_{\text{PT}}$ AVG | CODE$_{\text{PT}}$ AVG | GPR$_{\text{PT}}$ AVG |
|---|---|---|---|---|---|
| $\mathcal{M}_{\text{base}}$ | 52.70 | 47.17 | 47.13 | 40.89 | 75.63 |
| $\mathcal{M}_{\text{SHQ}}$ | 54.98 | 52.60 | 46.90 | 44.32 | 76.09 |
| $\mathcal{M}_{\text{LDQ}}$ | 64.09 | 75.56 | 54.38 | 49.94 | 76.48 |
| $\mathcal{M}_{\text{LMQ}}$ | 64.07 | 72.37 | 54.49 | 52.60 | 76.83 |
| $\mathcal{M}_{\text{res}}$ | **61.05** | 66.84 | 51.92 | 48.95 | 76.46 |

Table 1: **Average Accuracies of base models trained without or with varying $\mathcal{D}_{\text{res}}$.** Pretraining with diverse reasoning data yields immediate gains, with scale and diversity driving math and code improvements, more than quality. $\mathcal{M}_{\text{res}}$ represents the average of $\mathcal{M}_{\text{SHQ}}$, $\mathcal{M}_{\text{LDQ}}$, and $\mathcal{M}_{\text{LMQ}}$.

reasoning data surpasses $\mathcal{M}_{\text{base}}$. $\mathcal{M}_{\text{res}}$, average of the three reasoning-augmented variants trained under the same 1T token budget, improves over $\mathcal{M}_{\text{base}}$ by +8.35% on overall average accuracy. The largest improvements come from models trained on large-scale, diverse data; $\mathcal{M}_{\text{LDQ}}$ achieves

---

[2]https://github.com/NVIDIA/NeMo-Skills

highest average, driven by a +28.4% gain in mathematics and a +9% gain in code over the baseline. Interestingly, the smaller, less diverse, high-quality dataset ($\mathcal{M}_{\text{SHQ}}$) provides a modest lift, suggesting that at this early stage, the scale and diversity of the reasoning data are more critical than its curated quality for establishing a broad and robust reasoning foundation. Our experiments with a 1.2B Transformer (see Table 14) demonstrate that this front-loading strategy yields consistent, scalable performance gains—confirming the robustness of our approach across varying architectures.

**Pretraining Advantage is Maintained and Amplified Post-SFT.** We evaluate whether a strong SFT phase can close the accuracy gap established during pretraining with diverse reasoning data $\mathcal{D}_{\text{res}}$. At the same time, we examine whether the inclusion of such data causes the model to overfit and reduce generalization, thereby diminishing subsequent post-training gains. We finetune each pretrained model on three reasoning datasets ($\mathcal{D}_{\text{SHQ}}$, $\mathcal{D}_{\text{LDQ}}$, $\mathcal{D}_{\text{LMQ}}$), producing 12 models in total. We report the average results in Table 2 and include the full breakdown in Table 13.

| Model | Average | MATH$_{\text{SFT}}$ AVG | SCIENCE$_{\text{SFT}}$ AVG | CODE$_{\text{SFT}}$ AVG | INS$_{\text{SFT}}$ AVG |
|---|---|---|---|---|---|
| $\mathcal{M}_{\text{base}}$ + SFT | 26.62 | 34.48 | 20.92 | 7.09 | 43.98 |
| $\mathcal{M}_{\text{res}}$ + SFT | **35.92** | 40.61 | 34.77 | 16.75 | 51.52 |

Table 2: **Average Accuracies of SFT models pretrained with varying $\mathcal{D}_{\text{res}}$.** SFT amplifies the pretraining advantage—models with reasoning-rich pretraining significantly outperform baseline.

The results in Table 2 indicate that the advantage gained during the pre-training phase not only persists but is amplified. The group of models pretrained with reasoning data ($\mathcal{M}_{\text{res}}$ + SFT) outperforms the baseline group ($\mathcal{M}_{\text{base}}$ + SFT) by a significant 9.3% on average. This result strongly refutes the "catch-up" hypothesis, showing that SFT is not a substitute for a strong reasoning foundation built during pretraining. While recent works have found reasoning-centric post-training to be most effective on math domains, the improvement on science is minimal (Prabhakar et al., 2025; Luo et al., 2025a; Huan et al., 2025). However, the accuracy disparity in our findings is most prominent in science domains, an area often overlooked in reasoning-focused post-training work. This suggests that pretraining with reasoning data does more than teach facts; it helps the model develop effective internal representations for abstract and logical structures to enhance problem solving ability across domains. It does not overfit the model rather infuses the critical thinking ability that comes into full potential after post-training (Appendix B). Consequently, the model's capacity to absorb and leverage the SFT data is fundamentally enhanced, leading to greater learning efficiency and a higher performance ceiling. SFT acts as a powerful enhancer, but its ultimate effectiveness is constrained by the quality of the foundation established during pretraining.

| Model | Avg. | Math Reasoning | | | | Science & Code Reasoning | | | |
|---|---|---|---|---|---|---|---|---|---|
| | | MATH-500 | GSM8K | AIME24 | AIME25 | GPQA | MMLU | MMLU-PRO | LCB |
| $\mathcal{M}_{\text{base}}$ + SFT$_{\text{SHQ}}$ + RL | 37.92 | 72.05 | 83.83 | 12.29 | 16.04 | 28.16 | 41.10 | 36.69 | 13.16 |
| $\mathcal{M}_{\text{LMQ}}$ + SFT$_{\text{SHQ}}$ + RL | **56.66** | 87.13 | 93.07 | 45.21 | 33.96 | 31.69 | 72.91 | 56.91 | 32.43 |

Table 3: **Average accuracies of RL models pretrained and fine-tuned with varying $\mathcal{D}_{\text{res}}$.** Introducing reasoning data early provides significant reasoning boost after post-training.

**Pretraining Strategy Dictates Final Accuracy on Expert-Level Tasks.** The final RL phase reveals the definitive impact of our pretraining interventions, particularly on expert-level reasoning benchmarks. We select $\mathcal{M}_{\text{LMQ}}$ + SFT$_{\text{SHQ}}$ and $\mathcal{M}_{\text{base}}$ + SFT$_{\text{SHQ}}$ finetuned using $\mathcal{D}_{\text{SHQ}}$ as our two extreme pretraining backbones. As shown in Table 3, the accuracy gap between the two models continues to diverge, with the fully-aligned $\mathcal{M}_{\text{LMQ}}$ models achieving a 18.57% lead over the $\mathcal{M}_{\text{base}}$ model on average. The most striking results appear on the highly challenging AIME competition math problems, where the reasoning-pretrained models deliver a 39.32% improvement over the baseline. This provides conclusive evidence that early investment in reasoning data yields compounding returns, becoming the decisive factor in achieving frontier accuracies on the most demanding tasks.

## 5 ABLATIONS

**Does the scale and diversity of the reasoning data matter in Pretraining?** As detailed in Table 1, plainly increasing size and diversity of $\mathcal{D}_{\text{res}}$ in pretraining has significant improvement on

| Model | Average | MATH$_{\text{SFT}}$ AVG | SCIENCE$_{\text{SFT}}$ AVG | CODE$_{\text{SFT}}$ AVG | INS$_{\text{SFT}}$ AVG |
|---|---|---|---|---|---|
| $\mathcal{M}_{\text{base}} + \text{SFT}_{\text{SHQ}}$ | 29.92 | 42.79 | 35.83 | 10.48 | 30.59 |
| $\mathcal{M}_{\text{base}} + \text{SFT}_{\text{SHQ}}(2\times\text{epochs})$ | 34.01 | 48.05 | 40.69 | 14.60 | 32.70 |
| $\mathcal{M}_{\text{SHQ}} + \text{SFT}_{\text{SHQ}}$ | 37.33 | 50.52 | 40.00 | 24.76 | 34.06 |
| $\mathcal{M}_{\text{LDQ}} + \text{SFT}_{\text{SHQ}}$ | 46.70 | 60.79 | 50.67 | 28.57 | 46.79 |
| $\mathcal{M}_{\text{LMQ}} + \text{SFT}_{\text{SHQ}}$ | **50.95** | 64.67 | 53.74 | 35.55 | 49.82 |

Table 4: **Impact of diverse pretraining** $\mathcal{D}_{\text{res}}$ **on SFT phase.** Doubling SFT for the baseline fails to "catch up" to reasoning-pretrained models, while the latent advantage of the mixed-quality pretraining ($\mathcal{M}_{\text{LMQ}}$) emerges, making it the top performer.

the base model. The model pretrained on large, diverse data ($\mathcal{M}_{\text{LDQ}}$) achieves an absolute +9.09% average gain over the model trained on the smaller, less diverse corpus ($\mathcal{M}_{\text{SHQ}}$), with the largest gains observed in math, science, and code—domains that explicitly demand structured reasoning. GPR$_{\text{PT}}$ AVG shows limited sensitivity to diversity due to the nature of tasks that require commonsense and general knowledge. In contrast, scaling $\mathcal{D}_{\text{LDQ}}$ with $\mathcal{D}_{\text{SHQ}}$ (high-quality but less diverse) as in $\mathcal{M}_{\text{LMQ}}$ provides minimal further benefit on the reasoning tasks—underscoring that broad exposure to diverse reasoning patterns during pretraining is impactful for building a strong foundation.

***The Pretraining Advantage Persists and Resists "Catch-Up" Attempts via SFT.*** A central question is whether a model without a reasoning-rich pretraining ($\mathcal{M}_{\text{base}}$) can compensate for this deficit by undergoing a more intensive SFT phase. We test this "catch-up" hypothesis by fine-tuning $\mathcal{M}_{\text{base}}$ with two times more epochs using the same SFT data ($\mathcal{D}_{\text{SHQ}}$). The results in Table 4 prove this hypothesis false. While doubling the SFT epochs improves the baseline's average score by 4.09%, this enhanced baseline **still fails to match** the performance of even our weakest reasoning-pretrained model, $\mathcal{M}_{\text{SHQ}} + \text{SFT}_{\text{SHQ}}$ (+3.32%). This provides strong evidence that pretraining instills a foundational reasoning capability that cannot be fully replicated by simply scaling the SFT phase.

***Post-SFT, high-quality data reveals latent value.*** The downstream consequences of these pretraining choices become more nuanced after SFT. To isolate and test whether these effects persist into post-training, we finetune all base models with the same high-quality SFT recipe ($\mathcal{D}_{\text{SHQ}}$). Results in Table 4 confirm that models pretrained on diverse corpora continue to substantially outperform less diverse counterparts even after SFT, confirming that a diverse pretraining foundation enhances a model's capacity to benefit from SFT. More surprisingly, while the immediate gains of scaling with high-quality but narrow data ($\mathcal{M}_{\text{LMQ}}$) were muted at the pretraining stage, SFT reveals a latent advantage: $\mathcal{M}_{\text{LMQ}}$ achieves an additional +4.25% gain over $\mathcal{M}_{\text{LDQ}}$ post-SFT. This reveals a critical finding that high-quality but less diverse data may act as a *complementary amplifier*, whose benefits emerge after alignment—underlining the latent impact of quality of data during the pretraining.

| Model | Average | MATH$_{\text{SFT}}$ AVG | SCIENCE$_{\text{SFT}}$ AVG | CODE$_{\text{SFT}}$ AVG | INS$_{\text{SFT}}$ AVG |
|---|---|---|---|---|---|
| $\mathcal{M}_{\text{base}} + \text{SFT}_{\text{SHQ}}$ | 29.92 | 42.79 | 35.83 | 10.48 | 30.59 |
| $\mathcal{M}_{\text{res}} + \text{SFT}_{\text{LMQ}}$ | 31.21 | 30.91 | 27.73 | 9.79 | 56.41 |
| $\mathcal{M}_{\text{res}} + \text{SFT}_{\text{LDQ}}$ | 31.54 | 32.28 | 28.43 | 10.85 | 54.61 |
| $\mathcal{M}_{\text{res}} + \text{SFT}_{\text{SHQ}}$ | **44.99** | 58.66 | 48.14 | 29.63 | 43.56 |

Table 5: **Impact of diverse SFT** $\mathcal{D}_{\text{res}}$ **on SFT phase.** Fine-tuning on the small, high-quality corpus ($\mathcal{D}_{\text{SHQ}}$) is highly effective, while using large, diverse corpora ($\mathcal{D}_{\text{LDQ}}$) degrades reasoning.

***SFT is dominated by data quality, not diversity.*** We finetune all reasoning-pretrained models ($\mathcal{M}_{\text{res}}$) on each of our distinct reasoning datasets, and report the averaged results in Table 5. The findings reveal a striking contrast: while diversity is beneficial in pretraining, blindly scaling diverse reasoning data during SFT degrades performance. Models trained with $\mathcal{D}_{\text{LDQ}}$ or $\mathcal{D}_{\text{LMQ}}$ during SFT underperform relative to those finetuned on the smaller, high-quality, long-CoT dataset, $\mathcal{D}_{\text{SHQ}}$, despite having been exposed to reasoning data during pretraining. In fact, the use of large-scale, mixed-quality data at the SFT stage not only erodes the benefits of reasoning-rich pretraining but can even lead to worse outcomes than the baseline $\mathcal{M}_{\text{base}}$ finetuned with $\mathcal{D}_{\text{SHQ}}$ in math, code, and

| Model | $\mathcal{D}_{\text{base}} : \mathcal{D}_{\text{res}}$ | Overall | MATH_PT AVG | SCIENCE_PT AVG | CODE_PT AVG | GPR_PT AVG |
|---|---|---|---|---|---|---|
| $\mathcal{M}_{\text{LMQ}}$ | 80 / 20 | 64.07 | 72.37 | 54.49 | 52.60 | 76.83 |
| | 90 / 10 | 63.97 | 75.24 | 53.21 | 50.19 | 77.23 |
| | 60 / 40 | 67.28 | 79.63 | 55.73 | 56.47 | 77.31 |

Table 6: **Effect of varying reasoning data ratio during pretraining.** Increasing the reasoning proportion improves reasoning-focused benchmarks while preserving general-domain performance.

science tasks which benefit from reasoning. This result confirms the widely held view that data quality and long reasoning data is critical for effective SFT (Zhou et al., 2023; Zhao et al., 2024; Prabhakar et al., 2025). Our findings, however, extend this understanding by showing that simply applying high-quality data at every stage is not optimal. Instead, the most effective strategy is **asymmetric**: pretraining benefits most from broad and diverse reasoning data to establish generalizable priors, whereas SFT requires high-quality, reasoning-heavy data for targeted refinement.

**Reasoning Ratio Sensitivity and Its Interaction with Alignment.** As outlined in Section 2.3, the ratio between reasoning data and general pretraining data is fixed at 20% on 400B tokens in our main experiments to ensure fair comparison across $\mathcal{D}_{\text{SHQ}}$, $\mathcal{D}_{\text{LDQ}}$, and $\mathcal{D}_{\text{LMQ}}$. While this provides a controlled setting, the optimal ratio is inherently empirical and may vary across domains and datasets. To study the sensitivity of this hyperparameter, we conduct additional experiments by varying the reasoning proportion using $\mathcal{D}_{\text{res}} = \mathcal{D}_{\text{LMQ}}$.

As detailed in Table 6, increasing the reasoning ratio from 20% to 40% improves overall pretraining accuracy and substantially boosts math, science, and code benchmarks. Reducing it to 10% produces a mild drop in overall performance. These results indicate that the reasoning signal scales positively with its proportion during pretraining.

| Model | Overall | MATH_SFT AVG | SCIENCE_SFT AVG | CODE_SFT AVG | INS_SFT AVG |
|---|---|---|---|---|---|
| $\mathcal{M}_{\text{LMQ}} + \text{SFT}_{\text{SHQ}}$ | 50.95 | 64.67 | 53.74 | 35.55 | 49.82 |
| $\mathcal{M}_{\text{LMQ}} + \text{SFT}_{\text{SHQ}}$ [60/40] | 52.63 | 67.71 | 54.19 | 43.81 | 44.81 |

Table 7: **Effect of pretraining reasoning ratio on downstream SFT performance.** Higher reasoning proportion improves reasoning benchmarks but slightly reduces instruction-following metrics.

We further examine how these ratios affect downstream performance after SFT. Using $\mathcal{D}_{\text{SHQ}}$ for fine-tuning under the same SFT recipe, we compare the 80/20 and 60/40 pretrained models in Table 7. The 60/40 model continues to improve reasoning benchmarks after SFT, particularly in math and code, but shows a decline in instruction-following performance. This aligns with the breadth–alignment trade-off discussed in Table 17. Increasing reasoning proportion strengthens structured reasoning capabilities, while general pretraining data contributes broader stylistic and formatting exposure that supports instruction-following flexibility. In summary, these experiments indicate that the reasoning ratio is a meaningful control knob. Higher proportions consistently strengthen reasoning performance both before and after SFT, but may modestly reduce alignment-sensitive metrics. The optimal balance may therefore depend on the target deployment domain, and systematic exploration across datasets remains an important direction for future work.

**How should we expand reasoning data during SFT?** We next ablate the effect of scaling reasoning data during the SFT phase by contrasting two strategies: (i) scaling with data of *similar quality and diversity*, and (ii) scaling with data of *higher quality and reasoning depth*. As shown in Table 8, simply doubling the amount of diverse but mixed-quality data yields negligible improvement in average accuracy with a 4.92% drop in math accuracy—suggesting that increasing the volume of noisy or shallow reasoning data may dilute the useful signal and actively harm reasoning-specific domains. The small gains in science and code do not offset this regression, highlighting the limits of quantity-driven scaling in SFT. In contrast, when scaling $\mathcal{D}_{\text{ALF}}$ with high-quality $\mathcal{D}_{\text{SHQ}}$ ($\mathcal{D}'_{\text{ALF}}$), the average accuracy improves further, with math and instruction-following tasks benefiting most. Importantly, this qualitative expansion is achieved with only a marginal increase in dataset size (0.4% more samples). These contrasting outcomes provide clear evidence that SFT is a phase of targeted

| Model | Average | MATH$_\text{SFT}$ AVG | SCIENCE$_\text{SFT}$ AVG | CODE$_\text{SFT}$ AVG | INS$_\text{SFT}$ AVG |
|---|---|---|---|---|---|
| $\mathcal{M}_\text{LDQ} + \text{SFT}_\text{LDQ}$ | 32.84 | 28.38 | 35.22 | 10.16 | 57.61 |
| $\mathcal{M}_\text{LDQ} + \text{SFT}_\text{2×LDQ}$ | 32.99 | 23.46 | 39.65 | 11.75 | 57.10 |
| $\mathcal{M}_\text{LDQ} + \text{SFT}_\text{ALF}$ | 42.66 | 60.95 | 47.29 | 22.54 | 39.87 |
| $\mathcal{M}_\text{LDQ} + \text{SFT}_{\text{ALF}'}$ | 43.04 | 61.61 | 45.78 | 22.53 | 42.23 |

Table 8: **Impact of scaling reasoning data in SFT phase.** Naively doubling mixed-quality data is detrimental to math reasoning, whereas targeted scaling of high-quality data yields consistent gains.

refinement, not broad data absorption; the most effective scaling strategy is to strategically enhance the training corpus with high-quality, reasoning-intensive examples.

# 6 RELATED WORK

**Reasoning in Pretraining and Midtraining.** Cheng et al. (2024) study *instruction pretraining* by converting raw text into short QA pairs and report gains on general-purpose reasoning tasks that require minimal reasoning. While effective for broad linguistic alignment, their setup does not explicitly target reasoning-intensive domains such as mathematics, graduate level science, or code. Moreover, their pipeline of self-distilled instruction generation demonstrates that Instruct-PT outperforms vanilla PT after instruction tuning, but it does not assess whether these marginal pretraining gains persist once models undergo reasoning-heavy SFT and reinforcement learning. In contrast, our work systematically varies the *complexity, quantity, and diversity* of reasoning-style SFT data—containing intermediate thoughts and answers—across both pretraining and SFT, allowing us to probe whether early exposure yields durable downstream advantages.

More recent efforts have begun to explore the interplay between pretraining and instruction tuning. Liang et al. (2025) augment the instruction-tuning pool to better align with the distribution of pretraining data, reinforcing consistency between the two stages. While complementary in spirit, their method is applied only during SFT and does not address whether reasoning-specific supervision at the pretraining stage provides sustained benefits. Similarly, Wang et al. (2025); AI et al. (2025) introduce a *mid-training* phase, continuing pretraining on a small but high-quality reasoning dataset before SFT and RLVR. They report substantial downstream gains, particularly in mathematics benchmarks, highlighting the promise of mid-training interventions. However, because their corpus is heavily math-centric, it is difficult to disentangle whether the improvements stem from scale, complexity, or domain diversity, and the generalizability to science or code remains unclear.

A complementary direction is pursued by Gandhi et al. (2025), who inject algorithmically generated "cognitive behavioral" reasoning traces during mid-training, demonstrating improvements after reinforcement learning. This underscores the potential of early reasoning supervision but remains limited in scope: the interventions are restricted to small datasets and narrow tasks, leaving open questions about scalability, diversity, and phase-specific allocation of reasoning data. Our work builds on these insights by conducting the first systematic, large-scale analysis of reasoning data across both pretraining and SFT, providing a principled framework for understanding *when* and *how* reasoning supervision should be applied.

# 7 CONCLUSION

Our study provides the first systematic investigation of how reasoning data, varying in scale, diversity, and quality, influences LLMs across the entire training pipeline. We show that reasoning must be introduced early: front-loading into pretraining creates durable foundations that post-training alone cannot recover. Crucially, we uncover an asymmetric allocation principle—diversity drives pretraining effectiveness, while quality governs SFT—providing a clear, actionable blueprint for data strategy. Further, we demonstrate that high-quality pretraining data can yield latent benefits activated only during SFT, and that naive SFT scaling with noisy data can be actively harmful. Collectively, these findings challenge the conventional division between pretraining and reasoning, positioning reasoning-aware pretraining as a critical ingredient in building more capable, generalizable, and compute-efficient language models.

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

## A  EXPERIMENTS AND RESULTS

**Breakdown of Pretraining Results.** Table 9 provides a fine-grained view of the pretraining effects across individual benchmarks, complementing the domain-level averages reported in Table 1. The improvements are not confined to a small subset of tasks but are broadly distributed, with the largest gains concentrated in mathematically intensive and program synthesis benchmarks. For example, on GSM8K and MATH-500, models trained with large-scale reasoning data exhibit dramatic jumps over $\mathcal{M}_{\text{base}}$, with $\mathcal{M}_{\text{LDQ}}$ and $\mathcal{M}_{\text{LMQ}}$ more than doubling performance on MATH-500. Similar trends appear in code generation tasks such as HUMANEVAL, HUMANEVALPLUS, and MBPP, where reasoning-augmented models consistently outperform the baseline, indicating stronger procedural and compositional capabilities.

In contrast, gains on general-purpose reasoning benchmarks such as ARC-C, RACE, WINO-GRANDE, and HELLASWAG are more modest, suggesting that these tasks are less sensitive to explicit reasoning supervision during pretraining. Nevertheless, even in these cases, performance remains stable or slightly improved, indicating that incorporating reasoning data does not harm broad language understanding. Overall, the benchmark-level results reinforce the conclusion that early exposure to diverse reasoning corpora yields systematic improvements, with the most pronounced benefits emerging on tasks that require multi-step deduction, symbolic manipulation, or structured program synthesis.

**Detailed SFT Benchmark Performance.** While the Table 2 presents aggregate performance metrics across different reasoning domains, Table 10, 11, and 12 provide a granular breakdown of how each reasoning model performs across individual benchmarks when fine-tuned on $\mathcal{D}_{\text{SHQ}}$, $\mathcal{D}_{\text{LDQ}}$, and $\mathcal{D}_{\text{LMQ}}$ respectively.

These detailed results confirm that the "reasoning advantage"—established during the 1T pretraining phase—is not limited to aggregate scores but persists across all distinct evaluation categories, including AIME, GSM8K, MMLU, and LIVECODEBENCH. Models pretrained with reasoning-specific corpora ($\mathcal{M}_{\text{LDQ}}$ and $\mathcal{M}_{\text{LMQ}}$) consistently outperform $\mathcal{M}_{\text{base}}$ and $\mathcal{M}_{\text{SHQ}}$ across these benchmarks, regardless of the SFT dataset used. Specifically, $\mathcal{M}_{\text{LMQ}} + \text{SFT}$ frequently achieves the highest

| Benchmark | $\mathcal{M}_{\text{base}}$ | $\mathcal{M}_{\text{SHQ}}$ | $\mathcal{M}_{\text{LDQ}}$ | $\mathcal{M}_{\text{LMQ}}$ | $\mathcal{M}_{\text{res}}$ |
|---|---|---|---|---|---|
| ARC-C | 80.89 | 80.46 | **81.40** | **81.83** | **81.15** |
| RACE | 73.59 | 75.41 | **78.28** | **79.43** | **76.68** |
| WINOGRANDE | 70.64 | 71.43 | **69.53** | **69.38** | **70.25** |
| HELLASWAG | 77.38 | 77.06 | **76.69** | **76.67** | **76.95** |
| GSM8K | 59.74 | 65.20 | **82.71** | **85.14** | **73.20** |
| MATH-500 | 34.60 | 40.00 | **68.40** | **59.60** | **50.65** |
| MMLU | 61.67 | 61.45 | **65.87** | **65.42** | **63.60** |
| MMLU-PRO | 32.59 | 32.34 | **42.89** | **43.56** | **37.85** |
| HUMANEVAL | 37.44 | 41.04 | **48.63** | **51.68** | **44.70** |
| HUMANEVALPLUS | 32.59 | 35.03 | **42.74** | **46.28** | **39.16** |
| MBPP | 41.64 | 47.47 | **48.85** | **51.47** | **47.36** |
| MBPP[SANITIZED] | 51.87 | 53.74 | **59.53** | **60.97** | **56.53** |
| MATH$_{\text{PT}}$ AVG | 47.17 | 52.60 | **75.56** | **72.37** | **61.92** |
| SCIENCE$_{\text{PT}}$ AVG | 47.13 | 46.90 | **54.38** | **54.49** | **50.72** |
| CODE$_{\text{PT}}$ AVG | 40.89 | 44.32 | **49.94** | **52.60** | **46.94** |
| GPR$_{\text{PT}}$ AVG | 75.63 | 76.09 | **76.48** | **76.83** | **76.25** |
| **Overall** | 52.70 | 54.98 | **64.09** | **64.07** | **61.05** |

Table 9: Breakdown of base model accuracies across benchmarks. With increasing diversity and quality, the difference between $\mathcal{M}_{\text{base}}$ and models pretrained with reasoning data increases.

| SFT Dataset: $\mathcal{D}_{\text{SHQ}}$ | | | | | |
|---|---|---|---|---|---|
| Benchmark | $\mathcal{M}_{\text{base}} + \text{SFT}$ | $\mathcal{M}_{\text{SHQ}} + \text{SFT}$ | $\mathcal{M}_{\text{LDQ}} + \text{SFT}$ | $\mathcal{M}_{\text{LMQ}} + \text{SFT}$ | $\mathcal{M}_{\text{res}} + \text{SFT}$ |
| IFEVAL | 30.59 | 34.06 | 46.79 | 49.82 | 43.56 |
| AIME-24 | 8.12 | 18.33 | 35.21 | 41.88 | 31.81 |
| AIME-25 | 11.88 | 18.12 | 29.38 | 33.12 | 26.87 |
| GSM8K | 81.24 | 86.58 | 91.05 | 92.84 | 90.16 |
| MATH-500 | 69.90 | 79.05 | 87.50 | 90.85 | 85.80 |
| MMLU | 52.14 | 62.90 | 71.15 | 73.49 | 69.18 |
| MMLU-PRO | 39.45 | 48.63 | 53.45 | 55.54 | 52.54 |
| GPQA-DIAMOND | 15.91 | 8.46 | 27.40 | 32.20 | 22.69 |
| LIVECODEBENCH | 10.48 | 24.76 | 28.57 | 35.55 | 29.63 |
| MATH$_{\text{SFT}}$ AVG | 42.79 | 50.52 | 60.79 | 64.67 | 58.66 |
| SCIENCE$_{\text{SFT}}$ AVG | 35.83 | 40.00 | 50.67 | 53.74 | 48.14 |
| CODE$_{\text{SFT}}$ AVG | 10.48 | 24.76 | 28.57 | 35.55 | 29.63 |
| INS$_{\text{SFT}}$ AVG | 30.59 | 34.06 | 46.79 | 49.82 | 43.56 |
| **Overall** | 35.52 | 42.32 | 52.28 | 56.14 | 50.25 |

Table 10: Breakdown of model accuracies across benchmarks after training SFT phase on the $\mathcal{D}_{\text{SHQ}}$.

peak performance in complex reasoning tasks like AIME-24/25 and MATH-500, suggesting that high-quality, dense reasoning pretraining provides a more robust initialization that SFT can more effectively refine.

To facilitate a direct comparison of how different SFT strategies interact with varying pretraining foundations, Table 13 aggregates the results across all model-SFT combinations. The data underscores a clear synergy: the best overall performance is achieved when models are pretrained on diverse, high-quality reasoning data and subsequently fine-tuned on a compatible SFT dataset. This highlights that while SFT is crucial for aligning model behavior, the "foundation" built during pretraining serves as the primary determinant for the model's upper bound on complex reasoning capabilities. Even when given the same SFT budget, models lacking reasoning-specific pretraining

| | SFT Dataset: $\mathcal{D}_{\text{LDQ}}$ | | | | |
|---|---|---|---|---|---|
| **Benchmark** | $\mathcal{M}_{\text{base}}$ + SFT | $\mathcal{M}_{\text{SHQ}}$ + SFT | $\mathcal{M}_{\text{LDQ}}$ + SFT | $\mathcal{M}_{\text{LMQ}}$ + SFT | $\mathcal{M}_{\text{res}}$ + SFT |
| IFEVAL | 50.86 | 47.01 | 57.61 | 59.21 | 54.61 |
| AIME-24 | 1.15 | 2.50 | 6.37 | 4.90 | 4.59 |
| AIME-25 | 0.83 | 3.12 | 7.71 | 9.38 | 6.74 |
| GSM8K | 73.56 | 75.11 | 59.81 | 77.62 | 70.84 |
| MATH-500 | 46.70 | 44.98 | 39.63 | 56.28 | 46.96 |
| MMLU | 15.25 | 9.95 | 49.15 | 56.81 | 38.64 |
| MMLU-PRO | 16.26 | 14.24 | 30.50 | 33.51 | 26.08 |
| GPQA-DIAMOND | 8.97 | 7.39 | 26.01 | 28.35 | 20.58 |
| LIVECODEBENCH | 6.04 | 10.48 | 10.16 | 11.91 | 10.85 |
| MATH$_{\text{SFT}}$ AVG | 30.56 | 31.43 | 28.38 | 37.04 | 32.28 |
| SCIENCE$_{\text{SFT}}$ AVG | 13.49 | 10.52 | 35.22 | 39.55 | 28.43 |
| CODE$_{\text{SFT}}$ AVG | 6.04 | 10.48 | 10.16 | 11.91 | 10.85 |
| INS$_{\text{SFT}}$ AVG | 50.86 | 47.01 | 57.61 | 59.21 | 54.61 |
| **Overall** | 25.24 | 24.86 | 32.84 | 36.93 | 31.54 |

Table 11: Breakdown of model accuracies across benchmarks after training SFT phase on the $\mathcal{D}_{\text{LDQ}}$.

| | SFT Dataset: $\mathcal{D}_{\text{LMQ}}$ | | | | |
|---|---|---|---|---|---|
| **Benchmark** | $\mathcal{M}_{\text{base}}$ + SFT | $\mathcal{M}_{\text{SHQ}}$ + SFT | $\mathcal{M}_{\text{LDQ}}$ + SFT | $\mathcal{M}_{\text{LMQ}}$ + SFT | $\mathcal{M}_{\text{res}}$ + SFT |
| IFEVAL | 50.50 | 52.65 | 57.78 | 58.79 | 56.41 |
| AIME-24 | 1.25 | 3.13 | 8.23 | 4.69 | 5.35 |
| AIME-25 | 0.84 | 2.92 | 6.98 | 7.09 | 5.66 |
| GSM8K | 72.93 | 74.03 | 57.70 | 76.27 | 69.33 |
| MATH-500 | 45.33 | 42.18 | 36.93 | 50.75 | 43.28 |
| MMLU | 15.75 | 6.90 | 50.90 | 55.15 | 37.65 |
| MMLU-PRO | 15.57 | 13.18 | 32.09 | 33.37 | 26.21 |
| GPQA-DIAMOND | 8.97 | 4.87 | 23.17 | 29.99 | 19.34 |
| LIVECODEBENCH | 4.76 | 9.37 | 10.95 | 9.05 | 9.79 |
| MATH$_{\text{SFT}}$ AVG | 30.09 | 30.56 | 27.46 | 34.70 | 30.91 |
| SCIENCE$_{\text{SFT}}$ AVG | 13.43 | 8.31 | 35.39 | 39.50 | 27.73 |
| CODE$_{\text{SFT}}$ AVG | 4.76 | 9.37 | 10.95 | 9.05 | 9.79 |
| INS$_{\text{SFT}}$ AVG | 50.50 | 52.65 | 57.78 | 58.79 | 56.41 |
| **Overall** | 24.69 | 25.22 | 32.89 | 35.51 | 31.21 |

Table 12: Breakdown of model accuracies across benchmarks after training SFT phase on the $\mathcal{D}_{\text{LMQ}}$.

struggle to close the performance gap, particularly in challenging domains like code generation and advanced mathematics.

## B ADDITIONAL ABLATIONS

**Generalization and Robustness Across Model Scales and Architectures.** To ensure transparency and reproducibility, all pretraining, SFT, and RL datasets—including Common Crawl, Arxiv, Wikipedia, StackExchange, GitHub, OpenWebText, and OpenWebMath—are derived from fully open corpora. To validate the external validity of our data strategy and the effects of front-loading reasoning data across varying scales and architectures, we extended our experiments to a Transformer-based model $\mathcal{M}$ with 1.2B parameters trained on 125B tokens. We compared the baseline $\mathcal{M}_{\text{base}}$ against $\mathcal{M}_{\text{LMQ}}$, which integrates early reasoning exposure from $\mathcal{D}_{\text{LMQ}}$ while maintaining a constant token ratio between $\mathcal{D}_{\text{base}}$ and $\mathcal{D}_{\text{LMQ}}$.

| Model | MATH$_{SFT}$ AVG | SCIENCE$_{SFT}$ AVG | CODE$_{SFT}$ AVG | INS$_{SFT}$ AVG |
|---|---|---|---|---|
| $\mathcal{M}_{base}$ + SFT$_{SHQ}$ | 42.79 | 35.83 | 10.48 | 30.59 |
| $\mathcal{M}_{base}$ + SFT$_{LDQ}$ | 30.56 | 13.49 | 6.04 | 50.86 |
| $\mathcal{M}_{base}$ + SFT$_{LMQ}$ | 30.09 | 13.43 | 4.76 | 50.50 |
| $\mathcal{M}_{base}$ + SFT | 34.48 | 20.92 | 7.09 | 43.98 |
| $\mathcal{M}_{SHQ}$ + SFT$_{SHQ}$ | 50.52 | 40.00 | 24.76 | 34.06 |
| $\mathcal{M}_{LDQ}$ + SFT$_{SHQ}$ | 60.79 | 50.67 | 28.57 | 46.79 |
| $\mathcal{M}_{LMQ}$ + SFT$_{SHQ}$ | 64.67 | 53.74 | 35.55 | 49.82 |
| $\mathcal{M}_{SHQ}$ + SFT$_{LDQ}$ | 31.43 | 10.52 | 10.48 | 47.01 |
| $\mathcal{M}_{LDQ}$ + SFT$_{LDQ}$ | 28.38 | 35.22 | 10.16 | 57.61 |
| $\mathcal{M}_{LMQ}$ + SFT$_{LDQ}$ | 37.04 | 39.55 | 11.91 | 59.21 |
| $\mathcal{M}_{SHQ}$ + SFT$_{LMQ}$ | 30.56 | 8.31 | 9.37 | 52.65 |
| $\mathcal{M}_{LDQ}$ + SFT$_{LMQ}$ | 27.46 | 35.39 | 10.95 | 57.78 |
| $\mathcal{M}_{LMQ}$ + SFT$_{LMQ}$ | 34.70 | 39.50 | 9.05 | 58.79 |
| $\mathcal{M}_{res}$ + SFT | 40.62 | 34.77 | 16.75 | 51.52 |

Table 13: **Results of all SFT models with varying pretraining and SFT data**. Model pretrained with reasoning data obtains the highest gain after SFT phase of training.

| Model | MATH$_{PT}$ AVG | SCIENCE$_{PT}$ AVG | CODE$_{PT}$ AVG | GPR$_{PT}$ AVG | Overall Avg |
|---|---|---|---|---|---|
| $\mathcal{M}_{base}$ | 6.92 | 18.91 | 12.82 | 40.92 | 19.89 |
| $\mathcal{M}_{LMQ}$ | 16.39 | 21.50 | 23.44 | 41.69 | 25.75 |

Table 14: **Effect of early reasoning exposure on a 1.2B Transformer model.** The reasoning-augmented model improves substantially on math, science, and code benchmarks while maintaining general-domain performance.

As summarized in Table 14, the integration of reasoning data consistently enhances task-specific performance across different architectures. We observed substantial gains, specifically +9.47% in Math, +2.60% in Science, and +10.62% in Code benchmarks. Critically, these enhancements do not degrade generalized capabilities; performance on general benchmarks (GPR$_{PT}$ AVG) remains stable, indicating that early exposure to reasoning data does not perturb the model's fundamental language modeling objectives. These findings provide strong empirical evidence that our data strategy is robust and scalable, demonstrating that front-loading reasoning data is an effective mechanism for augmenting complex deduction capabilities regardless of the underlying model scale or architecture.

**Anatomy of high-quality reasoning data in SFT.** Our previous results establish that SFT benefits immensely from high-quality data, but what precisely constitutes "quality" remains unclear. In this ablation, we investigate a defining characteristic of such data: the depth and complexity of its reasoning traces. Specifically, we compare datasets that differ both in reasoning length and construction method. The high-quality corpus $\mathcal{D}_{SHQ}$ consists of answers generated by strong teacher models, characterized by long chain-of-thoughts with an average length exceeding 10k tokens. In contrast, $\mathcal{D}_{LDQ}$ provides reasoning data from diverse domains but with much shorter and noisier reasoning traces (average ~550 tokens). This distinction highlights a potential mechanism underlying quality: longer reasoning chains may serve as richer supervisory signals, encouraging models to internalize structured multi-step inference rather than surface-level heuristics.

To test this hypothesis, we extract from $\mathcal{D}_{LLQ}$ only the longest reasoning traces, creating a new dataset $\mathcal{D}_{ALF}$. Although it represents only ~2% of the original $\mathcal{D}_{LLQ}$ corpus, $\mathcal{D}_{ALF}$ is highly skewed toward domains with inherently deeper reasoning (75% math, with the remainder in science, code, and general reasoning). We then conduct SFT on top of the $\mathcal{M}_{llq}$ model using both $\mathcal{D}_{LLQ}$ (quantity and diversity) and $\mathcal{D}_{ALF}$ (length-filtered complexity).

As shown in Table 15, emphasizing depth in reasoning traces has a significant impact on downstream reasoning tasks. While finetuning with $\mathcal{D}_{LLQ}$ yields only modest improvements, switching to the

| Model | Average | MATH$_{\text{SFT}}$ AVG | SCIENCE$_{\text{SFT}}$ AVG | CODE$_{\text{SFT}}$ AVG | INS$_{\text{SFT}}$ AVG |
|---|---|---|---|---|---|
| $\mathcal{M}_{\text{LDQ}} + \text{SFT}_{\text{LDQ}}$ | 32.84 | 28.38 | 35.22 | 10.16 | 57.61 |
| $\mathcal{M}_{\text{LDQ}} + \text{SFT}_{\text{ALF}}$ | 42.71 | 60.95 | 47.50 | 22.54 | 39.87 |

Table 15: **Impact of depth in reasoning traces in data on SFT phase.** Model trained on longer CoT reasoning data outperforms the one trained on diverse reasoning traces.

50 times smaller, filtered by reasoning depth via answer length $\mathcal{D}_{ALF}$ boosts the overall score to 9.87%, with particularly strong gains in math, science and code. Interestingly, this comes at the cost of slightly reduced accuracy on instruction-following tasks, reflecting a trade-off between breadth and reasoning-specific depth. These results provide strong evidence that *longer chain-of-thought supervision is a critical marker of quality in* SFT *data*. Even when drawn from a noisy, large-scale corpus, selecting for reasoning depth alone can yield outsized improvements, making length-filtering a simple yet cost-effective heuristic for constructing impactful reasoning datasets for SFT phase.

**Data Redundancy Reinforces Foundational Skills, Not Overfitting.** A critical consideration in our two-phase approach is whether using the same reasoning data in both pretraining and SFT leads to catastrophic forgetting or brittle overfitting, a known concern in sequential fine-tuning (Luo et al., 2025b; Chen et al., 2025a).

Our results, shown in Figure 2, suggest this concern is unfounded and that the opposite is true: for reasoning, strategic redundancy is highly beneficial. The baseline model, $\mathcal{M}_{base}$, exposed to the high-quality $\mathcal{D}_{\text{SHQ}}$ data only during SFT, is the lowest performer across all categories. In contrast, $\mathcal{M}_{\text{SHQ}}$, which sees this same data in both phases, demonstrates a significant performance uplift, indicating that the second exposure reinforces rather than overwrites learning. We hypothesize this occurs because the two training phases serve different learning functions. During pretraining, the reasoning data is integrated slowly into the model's core representations alongside vast, diverse knowledge, forcing an internalization of abstract logical patterns.

The SFT phase then acts not as a new learning task, but as a powerful reinforcement signal on an already-prepared foundation. This benefit is amplified by a diverse pretraining context: the top-performing $\mathcal{M}_{\text{LMQ}}$ model leverages its broad exposure to various reasoning styles to most effectively capitalize on the repeated, high-quality signal from $\mathcal{D}_{\text{SHQ}}$. This suggests that data redundancy between pretraining and SFT should be viewed as a powerful mechanism for skill consolidation, where a diverse pretraining builds the capacity for reasoning and redundant SFT sharpens it.

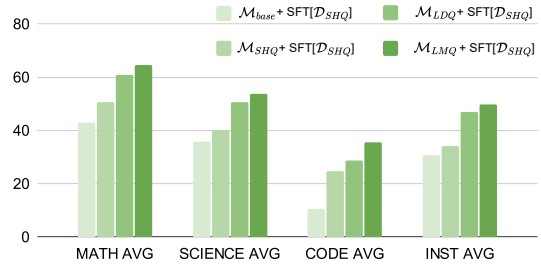

Figure 2: The model that saw the same high-quality data in both pretraining and SFT ($\mathcal{M}_{\text{SHQ}}$) handily beats the baseline ($\mathcal{M}_{base}$) that only saw the data once.

**Pretrained Foundations vs. Scaling SFT Data.** In Table 4, the notation $\text{SFT} \times 2$ refers to extending the SFT phase for twice the number of epochs over the existing dataset, rather than expanding the training set with unique samples. To further investigate whether reasoning performance is an artifact of data quantity—and to address potential concerns regarding overfitting with repeated data—we conducted a controlled experiment comparing a no-reason base model against our reason-aware baseline.We examined whether the no-reason base model, $\mathcal{M}_{\text{base}}$, could recover the performance of the reason-aware model, $\mathcal{M}_{\text{LDQ}}$, by utilizing a significantly larger volume of unique SFT tokens. We fine-tuned $\mathcal{M}_{\text{base}}$ on a combined dataset of $\mathcal{D}_{\text{LDQ}}$ and $\mathcal{D}_{\text{ALF}}$ (totaling 268M unique samples) and compared it against $\mathcal{M}_{\text{LDQ}}$, which was fine-tuned solely on $\mathcal{D}_{\text{ALF}}$ (7.1M unique samples). Both models were evaluated under the identical SFT evaluation setup.

| Model | MATH$_{\text{SFT}}$ AVG | SCIENCE$_{\text{SFT}}$ AVG | CODE$_{\text{SFT}}$ AVG | INS$_{\text{SFT}}$ AVG | Overall |
|---|---|---|---|---|---|
| $\mathcal{M}_{\text{base}} + \text{SFT}_{\text{LDQ+ALF}}$ | 33.66 | 29.15 | 3.49 | 56.86 | 30.79 |
| $\mathcal{M}_{\text{LDQ}} + \text{SFT}_{\text{ALF}}$ | 60.95 | 47.29 | 22.54 | 39.87 | 42.66 |

Table 16: **Comparison between SFT with large amounts of unique data and reasoning-augmented pretraining.** The reasoning-based model substantially outperforms the no-reason baseline despite using significantly fewer unique SFT samples.

| Instruction Type | $\mathcal{M}_{\text{LDQ}} + \text{SFT}_{\text{ALF}}$ | $\mathcal{M}_{\text{LDQ}} + \text{SFT}_{\text{LDQ}}$ | Diff |
|---|---|---|---|
| punctuation:no_comma | 13 | 27 | 14 |
| length_constraints:number_words | 19 | 30 | 11 |
| change_case:english_lowercase | 19 | 28 | 9 |
| keywords:letter_frequency | 13 | 21 | 8 |
| change_case:english_capital | 9 | 16 | 7 |
| language:response_language | 19 | 26 | 7 |
| detectable_format:number_bullet_lists | 14 | 21 | 7 |
| combination:two_responses | 11 | 16 | 5 |
| keywords:forbidden_words | 12 | 17 | 5 |
| detectable_format:title | 32 | 36 | 4 |
| startend:quotation | 19 | 23 | 4 |
| change_case:capital_word_frequency | 12 | 15 | 3 |
| length_constraints:number_paragraphs | 9 | 12 | 3 |
| length_constraints:nth_paragraph_first_word | 3 | 5 | 2 |
| length_constraints:number_sentences | 29 | 31 | 2 |
| detectable_format:json_format | 13 | 15 | 2 |
| startend:end_checker | 11 | 13 | 2 |
| detectable_format:number_highlighted_sections | 40 | 41 | 1 |
| detectable_format:constrained_response | 9 | 10 | 1 |
| detectable_content:number_placeholders | 23 | 23 | 0 |
| combination:repeat_prompt | 17 | 17 | 0 |
| detectable_format:multiple_sections | 12 | 12 | 0 |
| detectable_content:postscript | 23 | 22 | -1 |
| keywords:existence | 31 | 30 | -1 |
| keywords:frequency | 29 | 26 | -3 |

Table 17: Instruction-wise accuracy comparison on IFEval between models trained with less diverse ($\mathcal{D}_{\text{ALF}}$) and more diverse ($\mathcal{D}_{\text{LDQ}}$) corpora.

As demonstrated in Table 16, the reason-aware base model achieves a 39% relative improvement in the Overall score and consistently outperforms the no-reason baseline across all domains, despite the latter receiving substantially more unique training data. This result highlights that strong reasoning foundations established during pretraining cannot be trivially recovered through additional SFT, even when the model is provided with a significantly larger scale of unique data.

**Breadth vs. Alignment: The Role of Data Diversity in Instruction Following.** Table 5 reveals a consistent trade-off: length-filtered long-CoT SFT on $\mathcal{D}_{\text{ALF}}$ improves reasoning performance while weakening instruction-following ability. This pattern suggests that reduced data diversity affects alignment-sensitive behaviors. We hypothesize that this degradation stems from distributional skew and stylistic rigidity, as $\mathcal{D}_{\text{ALF}}$ is heavily concentrated in math and code domains, similar in structure to $\mathcal{D}_{\text{SHQ}}$. While such data reinforces precise token-level reasoning, it exposes the model to a narrower range of linguistic forms and formatting variations. To investigate this phenomenon, we compare instruction-level accuracy on the IFEval benchmark between two models: $\mathcal{M}_{\text{LDQ}} + \text{SFT}_{\text{ALF}}$ (less diverse SFT) and $\mathcal{M}_{\text{LDQ}} + \text{SFT}_{\text{LDQ}}$ (diverse SFT).

As shown in Table 17, the diverse $\mathcal{D}_{\text{LDQ}}$ dataset yields substantial improvements in linguistic manipulation tasks, including punctuation control, case transformation, word-count constraints, and formatting instructions. These categories require flexibility in natural language generation, sensitivity to stylistic variation, and the ability to adapt output structure. Exposure to diverse reasoning data

| Model | Overall Avg | MATH$_{PT}$ AVG | SCIENCE$_{PT}$ AVG | CODE$_{PT}$ AVG | GPR$_{PT}$ AVG |
|---|---|---|---|---|---|
| $\mathcal{M}_{LDQ}$ | 64.09 | 75.56 | 54.38 | 49.94 | 76.48 |
| $\mathcal{M}_{ALF}$ | 52.59 | 46.73 | 46.03 | 41.66 | 75.96 |

Table 18: **Effect of reducing scale and diversity of reasoning data during pretraining.** $\mathcal{M}_{ALF}$ is trained on a length-filtered subset of $\mathcal{D}_{LDQ}$ that is smaller and less diverse.

introduces broader linguistic cues, Markdown structures, and formatting styles, enabling stronger control over output form.

In contrast, the math- and code-heavy $D_{ALF}$ dataset reinforces strict token precision and pattern adherence. This benefits tasks involving keyword presence, token counting, or rigid structural constraints but provides limited stylistic variability. As a result, models trained on less diverse data excel at "hard" lexical constraints while underperforming on "soft" linguistic transformations that require adaptability. These findings clarify that data diversity plays a critical role in preserving instruction-following capabilities. Reasoning gains from length-filtered long-CoT data come with a narrowing of stylistic exposure, and broader linguistic coverage helps maintain alignment-related flexibility without sacrificing reasoning performance.

**Controlling for Scale and Diversity in Reasoning Pretraining.** To isolate the role of scale and diversity from other potential confounders such as dataset source or content differences, we conduct a controlled pretraining experiment using a downsampled subset of $\mathcal{D}_{LDQ}$. Specifically, we construct $\mathcal{D}_{ALF}$ by retaining only examples whose answer length exceeds 4096 tokens. This filtering procedure selects long-chain-of-thought samples that are typically math- and code-centric, resulting in data that is arguably high quality in reasoning depth but substantially smaller in scale and less diverse in topical coverage than the full $\mathcal{D}_{LDQ}$ corpus. During pretraining of $\mathcal{M}_{ALF}$, we maintain the same token ratio between $\mathcal{D}_{base}$ and $\mathcal{D}_{ALF}$ as in the original setup to ensure a controlled comparison.

As detailed in Table 18, the model trained on the downsampled subset exhibits an absolute 11.5 % drop in overall pretraining accuracy compared to $\mathcal{M}_{LDQ}$. The degradation is particularly pronounced in math and code benchmarks, while general-domain performance remains relatively stable. These results indicate that although long-chain-of-thought samples capture high-quality reasoning traces, reducing dataset scale and diversity substantially weakens the overall pretraining signal.

This controlled comparison supports the claim that reasoning quality alone is insufficient to explain performance gains. Instead, diversity and scale play a dominant role during pretraining, enabling broader generalization across reasoning-intensive domains.

