# OpenReview forum: "Front-Loading Reasoning: The Synergy between Pretraining and Post-Training Data"
_ICLR.cc/2026/Conference — ICLR 2026 Poster_

### Official Review · Reviewer_hBYq · 2025-10-20

**Soundness:** 3
**Presentation:** 2
**Contribution:** 2
**Rating:** 4
**Confidence:** 5

**Summary:**

This paper challenges the conventional practice of adding reasoning data only during fine-tuning. Through controlled experiments, it demonstrates that front-loading diverse reasoning data into pretraining creates an advantage that cannot be matched by later stages.

Their key findings are:

1. Pretraining benefits most from scale and diversity of reasoning data, building broad and robust foundational capabilities.
2. SFT is dominated by data quality, where small amounts of high-quality, long chain-of-thought data are vastly more effective than large, diverse but lower-quality mixtures. Naively scaling SFT data with mixed quality can even be detrimental.
3. High-quality pretraining data has a latent effect, where its full benefit is "unlocked" only after the subsequent SFT phase.

The study concludes that a strategic, asymmetric data strategy—prioritizing diversity in pretraining and quality in SFT—is essential for building more capable and efficient reasoning models.

**Strengths:**

1. The authors present a systematic and thorough experimental setup, including baseline comparisons, multiple data regimes (scale, quality, diversity), and evaluation across a broad suite of reasoning benchmarks and domains (math, science, code, general reasoning). This breadth of empirical evidence is a clear strength.
2. The central finding—that diversity in reasoning data benefits pretraining most, while data quality dominates SFT effectiveness—is supported by clear quantitative results (e.g., Tables 1, 2, and 3), providing a practical guideline for future LLM training recipes.
3. The paper performs extensive ablation studies (Section 5), specifically investigating scaling strategies, catch-up hypotheses, and the latent effects of high-quality data. For example, Table 4 rigorously compares the effects of doubling SFT data versus diverse pretraining.

**Weaknesses:**

1. The description of data volume is highly unclear. In Section 2.2, $D_{LDQ}$ is described as containing 336B tokens, while $D_{SHQ}$ is described as containing 1.2M samples. It is also unclear exactly how much base data and reasoning data each model used during the pretraining phase, especially for the $M_{SHQ}$ model.
2. The results in Tables 2, 3, and 5 only report the average scores of the three models pretrained with reasoning data. However, Table 1 has already demonstrated significant performance differences between the base models of LDQ and SHQ. The authors should provide the complete results for each individual model group across all tables.
3. The experimental results in this work lack validation across different model scales and on the more widely adopted Transformer architecture.

**Questions:**

See weakness.

---

> ### Author Response · Authors · 2025-11-20
> **Thank you for providing a constructive and detailed feedback!**
>
> We thank the reviewer for the thoughtful feedback and encouraging assessment of our work. We are glad that the systematic design of our experiments, the breadth of our evaluations, and the clarity of our main findings were evident. Your recognition of the practical value of our conclusions, as well as the depth of our ablation studies, is greatly appreciated. We look forward to addressing any further questions during the discussion period and will incorporate the feedback into the final manuscript.
>
> &nbsp;
>
> ---
>
> **- The description of the data volume is highly unclear. It is also unclear exactly how much base data and reasoning data each model used during the pretraining phase.**
>
> We thank the reviewer for pointing out the inconsistencies regarding data volume. We have revised the manuscript  (Line 158, 167-168) to explicitly state that $\mathcal{D}\_{\mathrm{LDQ}}$ contains 268M samples where $\mathcal{D}\_{\mathrm{SHQ}}$ consists of 1.2M samples and $\mathcal{D}\_{\mathrm{LMQ}}$ is the concatenation of $\mathcal{D}\_{\mathrm{LDQ}}$ and $\mathcal{D}\_{\mathrm{SHQ}}$ comprising 269M samples.
>
> Across all models, we keep the token ratio between $\mathcal{D}\_{\mathrm{base}}$ and $\mathcal{D}\_{\mathrm{res}}$ fixed during pretraining. When a reasoning dataset is small, it is repeated so that the model still observes the same total volume of reasoning tokens. To correctly state the data distribution, we pretrain all models for 600B tokens using $\mathcal{D}\_{\mathrm{base}}$ followed by 400B tokens on a mixture of 80% $\mathcal{D}\_{\mathrm{base}}$ and 20% $\mathcal{D}\_{\mathrm{res}}$. This results in a constant budget of 80B reasoning tokens across all experiments. $\mathcal{D}\_{\mathrm{res}}$ can be any one source of data among the three reasoning datasets we have defined in Section 2.2 ($\mathcal{D}\_{\mathrm{SHQ}}$, $\mathcal{D}\_{\mathrm{LDQ}}$, $\mathcal{D}\_{\mathrm{LMQ}}$). This token ratio has been maintained across all three pretraining runs with reasoning data.
>
> We highly appreciate the pretraining data distribution comment and acknowledge the incompleteness in the details. We have revised Section 2.2 and Appendix A to reflect these details accurately and explicitly.
>
> &nbsp;
>
> ---
> **- The authors should provide the complete results for each individual model group across all tables.**
>
> Thank you for pointing this out, and we have updated the individual results in Appendix (Tables 7,8,9,10) for brevity of space in the main paper. We have also added Table 11 to reflect the composition of Table 2 in detail.
>
> &nbsp;
>
> ---
>
> **- The experimental results in this work lack validation across different model scales and on the more widely adopted Transformer architecture.**
>
> We thank the reviewer for the feedback. To validate the transferability of our findings across different model scales and architectures, we pretrain a $\mathcal{M}$=1.2B transformer model for 125B tokens and observe the effect of early reasoning exposure. While pretraining $\mathcal{M}\_{\mathrm{LMQ}}$, we maintain the same token ratio between $\mathcal{D}\_{\mathrm{base}}$ and $\mathcal{D}\_{\mathrm{LMQ}}$ as the previous training setup. Finally, we evaluate both models on the same evaluation tasks as mentioned in the paper.
>
> &nbsp;
>
> | Model                         | AVG_MATH | AVG_SCIENCE | AVG_CODE | AVG_GENERAL | Overall Avg |
> |-------------------------------|----------|-------------|----------|-------------|-------------|
> | $\mathcal{M}\_{\mathrm{base}}$    | 6.92     | 18.91       | 12.82    | 40.92       | 19.89       |
> | $\mathcal{M}\_{\mathrm{LMQ}}$       | 16.39    | 21.50       | 23.44    | 41.69       | 25.75       |
>
> &nbsp;
>
> As shown in the table above, even with a transformer model (different architecture), the findings hold for a 1.2B model (different scale) trained for a much smaller token horizon (different training token budget). The gains of models trained with reasoning data is significantly higher specifically in reasoning tasks such as math (+9.47%), science (+2.60%) and code (+10.62%) tasks. Even inclusion of reasoning data does not perturb the performance of non-reasoning benchmarks (AVG_GENERAL). These results serve as strong empirical evidence that the proposed data strategy is effective across varying model scales and the widely adopted Transformer architecture.

---

> > ### Author Response · Authors · 2025-11-25
> > **We appreciate the feedback and would be happy to provide additional details!**
> >
> > Thank you for your detailed and thoughtful feedback! We hope the clarifications and additional experiments address your concerns and provide a comprehensive understanding of our work. Please let us know if there are any aspects that remain unclear or require further elaboration—we would be happy to provide additional details during this discussion period. We truly appreciate your valuable insights and hope these responses encourage you to reassess the current score where appropriate.

---

> > > ### Comment · Reviewer_hBYq · 2025-11-26
> > >
> > > Thanks to the authors for their answers. My concerns are all addressed, and I will raise my score.

---

> > > > ### Author Response · Authors · 2025-11-26
> > > > **Thank you for the reassessment!**
> > > >
> > > > We are delighted to hear that our responses successfully addressed your concerns. We sincerely appreciate your decision to raise the score and thank you for the constructive feedback that has helped improve our manuscript.

---

### Official Review · Reviewer_WLmB · 2025-10-23

**Soundness:** 2
**Presentation:** 2
**Contribution:** 3
**Rating:** 4
**Confidence:** 4

**Summary:**

This paper investigates the stage to introduce reasoning data into the training process of Large Language Models (LLMs). The authors challenge the conventional approach of adding reasoning skills during the post-training stage and  argue for incorporating it early on, during pretraining.

**Strengths:**

1. This paper provides a detailed comparison of pretraining with $\mathcal{M_{LDQ}}$, $\mathcal{M_{SHQ}}$, and a mixture of both, thoroughly evaluating their impact on the model's performance across different reasoning tasks after pretrain and SFT.
2. This paper shows that introducing reasoning data in pretrain stage can improve model performance in both pretrain stage and SFT stage
3. This paper demonstrates that the SFT stage should leverage higher-quality data with greater reasoning depth to significantly enhance the model's final performance.

**Weaknesses:**

1. All the performances are evaluated in reasoning tasks, the performance on the memry task are not evaluated.
2. The paper does not clearly specify how the $\mathcal{M_{LDQ}}$ and $\mathcal{M_{SHQ}}$ datasets were mixed to create $\mathcal{M_{LMQ}}$ (especially ratio). It would be important to investigate whether different mixing ratios significantly impact the model's performance, and specifically, whether an optimal ratio exists that would allow $\mathcal{M_{LMQ}}$ to outperform $\mathcal{M_{LDQ}}$.

**Questions:**

1. The objective in Equation 1 requires clarification. The notation $E_{t \sim \mathcal{T}}$ is ambiguous as to whether a single fine-tuned model is evaluated across all tasks in $\mathcal{T}$, or if a unique model is fine-tuned specifically for each task $t$.
2. Why choosing a hybrid model, is the conclusion the same for Decoder-only transformer?

---

> ### Author Response · Authors · 2025-11-20
> **Thank you for resonating with the strengths of our work and the detailed and insightful feedback!**
>
> We thank the reviewer for the positive and insightful feedback! We are encouraged that you recognized the significant contribution of our work, particularly our challenge to the conventional approach by front-loading reasoning data. We appreciate your positive remarks on our thorough evaluation of pretraining impact and our insights into the role of data quality during SFT. We believe all your questions can be addressed within this discussion period, but we would love to provide further clarification if needed and will incorporate the feedback in our final manuscript.
>
> &nbsp;
>
> -----
>
> **- All the performances are evaluated in reasoning tasks, the performance on the memry task are not evaluated.**
> &nbsp;
>
> Thank you for the suggestion. Along with the reasoning tasks, we have also evaluated the commonsense understanding of the base models across ARC Challenge, Winogrande, RACE and HellaSwag tasks. These tasks evaluate a model’s ability in language understanding, reading comprehension, and commonsense reasoning. We have provided a breakdown of each model performance across these tasks on Table 7 (also shown below) of the updated manuscript. As we can see from the results, we observe substantial improvements in reasoning tasks while maintaining stability in general benchmarks. This covers a large section of the tasks commonly used to assess base LLM performance. If the reviewer has specific tasks that they would like us to include, then please point us to the exact tasks and we will try our best to evaluate in the discussion time period.
>
>
> &nbsp;
>
> | Benchmark     | $\mathcal{M}\_{\mathrm{base}}$ | $\mathcal{M}\_{\mathrm{SHQ}}$ | $\mathcal{M}\_{\mathrm{LDQ}}$ | $\mathcal{M}\_{\mathrm{LMQ}}$ | $\mathcal{M}\_{\mathrm{res}}$ |
> |--------------|---------:|-------:|-------:|-------:|-------:|
> | ARC-C        | 80.89 | 80.46 | 81.40 | 81.83 | 81.15 |
> | RACE         | 73.59 | 75.41 | 78.28 | 79.43 | 76.68 |
> | WinoGrande   | 70.64 | 71.43 | 69.53 | 69.38 | 70.25 |
> | HellaSwag    | 77.38 | 77.06 | 76.69 | 76.67 | 76.95 |
>
> &nbsp;
>
> **N.B.,** $\mathcal{M}\_{\mathrm{res}} = Average(\mathcal{M}\_{\mathrm{SHQ}}, \mathcal{M}\_{\mathrm{LDQ}}, \mathcal{M}\_{\mathrm{LMQ}})$
>
> &nbsp;
>
> ----
>
> **- The paper does not clearly specify how the M_LDQ and  M_SHQ datasets were mixed to create M_LMQ (especially ratio).**
>
>
> Thank you for the feedback. We have revised the manuscript (Line 158, 167-168) to explicitly state that $\mathcal{D}\_{\mathrm{LDQ}}$ contains 268M samples and $\mathcal{D}\_{\mathrm{SHQ}}$ consists of 1.2M samples. $\mathcal{D}\_{\mathrm{LMQ}}$ is the concatenation of $\mathcal{D}\_{\mathrm{LDQ}}$ and $\mathcal{D}\_{\mathrm{SHQ}}$ comprising 269M samples. To summarize, we simply concatenate two datasets to obtain $\mathcal{D}\_{\mathrm{LMQ}}$.
>
> &nbsp;
>
> ----
>
> **- The objective in Equation 1 requires clarification.**
>
> We appreciate the reviewer’s request for clarification. In Equation 1, the expectation $\mathbb{E}\_{t \sim \mathcal{T}}$ defines a single fine-tuned model that is evaluated across the set of diverse tasks $\mathcal{T}$. We DO NOT fine-tune separate models for each individual task. Instead, each data configuration $(\mathcal{D}\_\mathrm{res}^\mathrm{PT},\mathcal{D}\_\mathrm{res}^\mathrm{SFT})$ produces one model, and that model is then evaluated on every benchmark task in $\mathcal{T}$. The expectation indicates task averaging in the reporting, not task-specific fine-tuning.
>
> We thank the reviewer for the feedback and have incorporated this clarification in the updated manuscript (Line 129-130).

---

> ### Author Response · Authors · 2025-11-20
> **Response to the questions**
>
> **- Why choosing a hybrid model, is the conclusion the same for Decoder-only transformer?**
>
> We thank the reviewer for the insightful feedback. We selected a hybrid model because of the growing interest in this design within the LLM community, driven by its cost-efficient inference [1]. However, to validate the transferability of our finding across different model scales and architectures, we pretrain a $\mathcal{M}$=1.2B transformer model for 125B tokens and observe the effect of early reasoning exposure. While pretraining $\mathcal{M}\_{\mathrm{LMQ}}$, we maintain the same token ratio between $\mathcal{D}\_{\mathrm{base}}$ and $\mathcal{D}\_{\mathrm{LMQ}}$ as the previous training setup. Finally, we evaluate both models on the same evaluation tasks as mentioned in the paper.
>
> &nbsp;
>
> | Model                         | MATH$_{\mathrm{PT}}$ AVG | SCIENCE$_{\mathrm{PT}}$ AVG | CODE$_{\mathrm{PT}}$ AVG | GPR$_{\mathrm{PT}}$ AVG | Overall Avg |
> |-------------------------------|:----------:|:-------------:|:----------:|:-------------:|:-------------:|
> | $\mathcal{M}\_{\mathrm{base}}$    | 6.92     | 18.91       | 12.82    | 40.92       | 19.89       |
> | $\mathcal{M}\_{\mathrm{LMQ}}$       | 16.39    | 21.50       | 23.44    | 41.69       | 25.75       |
>
> &nbsp;
>
> As shown in the table above, even with a transformer model (different architecture), the findings hold for a 1.2B model (different scale) trained for a much smaller token horizon (different training token budget). The gains of models trained with reasoning data are significantly higher, specifically in reasoning tasks such as math (+9.47%), science (+2.60%), and code (+10.62%) tasks. Even the inclusion of reasoning data does not perturb the performance of non-reasoning benchmarks (GPR$_{\mathrm{PT}}$ AVG). These results serve as strong empirical evidence that the proposed data strategy is effective across varying model scales and the widely adopted Transformer architecture. We are grateful to the reviewer for this insightful suggestion. We will definitely include this result in our paper to make our findings stronger and generalizable.
>
> [1] Blakeman, Aaron, et al. "Nemotron-h: A family of accurate and efficient hybrid mamba-transformer models." arXiv preprint arXiv:2504.03624 (2025).

---

> > ### Author Response · Authors · 2025-11-25
> > **We appreciate the feedback and would be happy to provide additional details!**
> >
> > Thank you once again for your thoughtful and constructive feedback. We have tried to carefully address all of your points in this discussion and will incorporate the necessary changes in the paper. We would greatly appreciate it if you could reconsider your score, given all the clarifications. However, if there are any further aspects or additional clarifications that you would like us to elaborate on, we would be more than happy to provide them.

---

> > > ### Comment · Reviewer_WLmB · 2025-11-26
> > >
> > > I will raise my score due to the concerns are solved in rebuttal stage.

---

> > > > ### Author Response · Authors · 2025-11-26
> > > > **Thank you for the reassessment!**
> > > >
> > > > We are glad to address the concerns and grateful to the reviewer for their time and invaluable feedback, which has helped make our paper stronger!

---

### Official Review · Reviewer_2v5m · 2025-10-27

**Soundness:** 3
**Presentation:** 3
**Contribution:** 3
**Rating:** 6
**Confidence:** 4

**Summary:**

This paper investigates the benefits of adding reasoning data earlier during pretraining. The authors vary the scale, diversity and quality of the reasoning data and show that front-loading reasoning data into pretraining stage not only builds better pretrained models but also synergies with subsequent post-training. They also show that diversity of the reasoning data is crucial for pretraining, whereas quality is important for SFT stage.

**Strengths:**

- The paper systematically studies the effects of front-loading different types of reasoning data (in terms of quality and diversity) into pretraining, which is novel and timely.

- The experiments are conducted at a large-scale and evaluated on diverse downstream tasks, which makes the conclusions more practical.

- The finding that high-quality pretraining data shows benefits only after SFT is interesting.

**Weaknesses:**

1. Major: 'Scale and diversity of the reasoning data are important in pretraining' is not well supported. D_{LDQ} and D_{SHQ} are from different datasets, and scale and diversity is not the only difference. The content of the dataset, as well as the quality of the dataset, are likely to be confounders. I think a more controllable comparison is to reduce the size and diversity of D_{LDQ} (-->D'_{LDQ}) and compare M_{LDQ} and M'_{LDQ}.

2. Minor: Some observations remain to be better understood, for example why exactly pretraining benefits more from diversity and SFT betnefits more from quality.

**Questions:**

1. Is D_{res} always 20% of the data (mentioned at Line 189)? If yes, why there are different scales of reasoning datasets (D_{SHQ} and D_{LDQ})? Can you clarify?

2. How is the ratio of reasoning data decided? Did you try other ratio (higher or lower than 20%) and does that affect your conclusions?

3. What is D_{LLQ}? at Line 171-172? It seems not defined.

4. Line 349-350, could you explain why D_{SHQ} is mentioned, should it be D_{LMQ}?

---

> ### Author Response · Authors · 2025-11-27
> **Thank you for the encouraging reviews and constructive feedbacks!**
>
> We are grateful for the reviewer’s positive feedback and support. We are glad to hear that our findings on front-loading reasoning data and the "latent" effects of pre-training were well-received. We are committed to polishing the work further and will update the paper accordingly to address all points raised during the review process.
>
> &nbsp;
>
> ---
> **- I think a more controllable comparison is to reduce the size and diversity of $\mathcal{D}\_{\mathrm{LLQ}}$ (-->$\mathcal{D}\_{\mathrm{LDQ}'}$) and compare $\mathcal{M}\_{\mathrm{LLQ}}$ and $\mathcal{M}\_{\mathrm{LLQ}'}$.**
>
> To address this concern, we conduct a controlled experiment by pretraining a model on downsampled subset of $\mathcal{D}\_{\mathrm{LDQ}}$, denoted $\mathcal{D}\_{\mathrm{ALF}}$, by retaining examples where the answer length exceeds 4096 tokens. This subset represents data that requires longer reasoning traces (more math and code centric data), thus of higher quality but significantly smaller in scale and less diverse than the full $\mathcal{D}\_{\mathrm{LDQ}}$. While pretraining $\mathcal{M}\_{\mathrm{ALF}}$, we maintain the same token ratio between $\mathcal{D}\_{\mathrm{base}}$ and $\mathcal{D}\_{\mathrm{ALF}}$ as the previous training setup.
>
> &nbsp;
>
>   | Model        | Overall | MATH$_{\mathrm{PT}}$ AVG | SCIENCE$_{\mathrm{PT}}$ AVG | CODE$_{\mathrm{PT}}$ AVG | GPR$_{\mathrm{PT}}$ AVG |
> |--------------|:-------------:|:----------:|:-------------:|:----------:|:-------------:|
> | $\mathcal{M}\_{\mathrm{LDQ}}$   | 64.09       | 75.56    | 54.38       | 49.94    | 76.48       |
> | $\mathcal{M}\_{\mathrm{ALF}}$   | 52.59       | 46.73    | 46.03       | 41.66    | 75.96       |
>
> &nbsp;
>
> The model trained with downsampled data shows an 11.5% drop in overall accuracy, indicating that reducing both scale and diversity leads to weaker pretraining performance. This validates our paper's central claim: while quality is crucial, diversity and scale are the dominant factors in the pre-training stage. We highly appreciate the valuable feedback. We will include the result in our final draft, which further strengthens our claim.
>
> &nbsp;
>
> ---
> **- Minor: Some observations remain to be better understood, for example why exactly pretraining benefits more from diversity and SFT benefits more from quality.**
>
> This is an excellent point. We postulate that Pre-training benefits from Diversity because its objective is to maximize the likelihood of the broad data distribution, ensuring the model does not suffer from mode collapse or "blind spots" in its reasoning capabilities.
>
> Conversely, SFT benefits from Quality because it is essentially a distribution-sharpening process (Table 22 of [1]). In this stage, the model does not need to learn new reasoning primitives (which it already possesses from pretraining), but rather needs to learn to consistently trigger its best reasoning path. High-quality data acts as a precise vector for this steering, whereas diverse/mixed-quality data dilutes the signal, leading to the regression we observed when scaling SFT blindly. We keep this discussion open for future work.
>
> [1] Guha, Etash, et al. "OpenThoughts: Data Recipes for Reasoning Models." arXiv preprint arXiv:2506.04178 (2025).

---

> > ### Author Response · Authors · 2025-11-27
> > **Response to the rebuttal [Part 2]**
> >
> > &nbsp;
> >
> > **- Is $\mathcal{D}\_{\mathrm{res}}$ always 20% of the data (mentioned at Line 189)? If yes, why are there different scales of reasoning datasets ($\mathcal{D}\_{\mathrm{SHQ}}$ and $\mathcal{D}\_{\mathrm{LDQ}}$)? Can you clarify?**
> >
> > We appreciate the clarification question. Across all models, we keep the token ratio between $\mathcal{D}\_{\mathrm{base}}$ and $\mathcal{D}\_{\mathrm{res}}$ fixed during pretraining. To correctly state the data distribution, we pretrain all models for 600B tokens using $\mathcal{D}\_{\mathrm{base}}$ followed by 400B tokens on a mixture of 80% $\mathcal{D}\_{\mathrm{base}}$ and 20% $\mathcal{D}\_{\mathrm{res}}$. This results in a constant budget of 80B reasoning tokens across all experiments. This implies that when a reasoning dataset is small, it is repeated so that the model still observes the same total volume of reasoning tokens during pretraining.
> >
> > We have revised the manuscript  (Line 158, 167-168) to explicitly state that $\mathcal{D}\_{\mathrm{LDQ}}$ contains 268M samples where $\mathcal{D}\_{\mathrm{SHQ}}$ consists of 1.2M samples and $\mathcal{D}\_{\mathrm{LMQ}}$ is the concatenation of $\mathcal{D}\_{\mathrm{LDQ}}$ and $\mathcal{D}\_{\mathrm{SHQ}}$ comprising 269M samples. The difference in sample sizes across reasoning datasets has been deliberately designed to pinpoint the effect of data towards each axis (1. Small-Scale, High-Quality Data, 2. Large-Scale, Mixed-Quality Data, and 3. Large-Scale, Diverse Data) during pretraining.
> > We highly appreciate the pretraining data distribution comment. We have revised Section 2.2 and Appendix A to reflect these details accurately and explicitly.
> >
> > &nbsp;
> >
> > ---
> > **- How is the ratio of reasoning data decided? Did you try other ratio (higher or lower than 20%) and does that affect your conclusions?**
> >
> > Thank you for raising this interesting point. In the ideal case, the ratio between reasoning data and general pretraining data should be determined empirically. In our experiments, we fix it at 20% for all reasoning datasets ($\mathcal{D}\_{\mathrm{SHQ}}$, $\mathcal{D}\_{\mathrm{LDQ}}$, $\mathcal{D}\_{\mathrm{LMQ}}$)  to ensure a fair comparison across base models. To address the reviewer’s question, we conduct a preliminary study with two additional ratios, using $\mathcal{D}\_{\mathrm{LMQ}}$ as reasoning data. The results are shown below:
> >
> > | Model                     | $\mathcal{D}\_{\mathrm{base}}$ Vs $\mathcal{D}\_{\mathrm{res}}$ | Overall | MATH$_{\mathrm{PT}}$ AVG | SCIENCE$_{\mathrm{PT}}$ AVG | CODE$_{\mathrm{PT}}$ AVG | GPR$_{\mathrm{PT}}$ AVG |
> > |---------------------------|:-----------------:|:-------------:|:-----------:|:--------------:|:-----------:|:--------------:|
> > | $\mathcal{M}_{\mathrm{LMQ}}$ | 80/20           | 64.07       | 72.37     | 54.49        | 52.60     | 76.83        |
> > | $\mathcal{M}_{\mathrm{LMQ}}$                          | 90/10           | 63.97       | 75.24     | 53.21        | 50.19     | 77.23        |
> > | $\mathcal{M}_{\mathrm{LMQ}}$                          | 60/40           | 67.28       | 79.63     | 55.73        | 56.47     | 77.31        |
> >
> > As shown in the table, increasing the ratio of reasoning data further increases the overall accuracy, while decreasing it results in a mild reduction. We further examined how this adjustment affects downstream performance after the SFT stage. Using $\mathcal{D}_{\mathrm{SHQ}}$ as the SFT dataset, we fine-tuned both the 80/20 and 60/40 pretrained models under the same SFT recipe used in the paper:
> >
> >
> > | Model                     | Overall | MATH$_{\mathrm{SFT}}$ AVG | SCIENCE$_{\mathrm{SFT}}$ AVG | CODE$_{\mathrm{SFT}}$ AVG | INST$_{\mathrm{SFT}}$ AVG |
> > |---------------------------|:---------:|:----------:|:--------------:|:----------:|:-----------:|
> > | $\mathcal{M}_{\mathrm{LMQ}} + \mathrm{SFT}\_{\mathrm{SHQ}}$           | 50.95   | 64.67    | 53.74        | 35.55    | 49.82     |
> > | $\mathcal{M}_{\mathrm{LMQ}} + \mathrm{SFT}\_{\mathrm{SHQ}}$[60-40]    | 52.63   | 67.71    | 54.19        | 43.81    | 44.81     |
> >
> > The result above clearly demonstrates that adding more reasoning data during pretraining continues to boost reasoning performance after SFT, though it introduces a decline in instruction-following metrics (IFEval). This is due to the fact that general pretraining data exposes the model to a wider range of writing styles, formats, and Markdown structures, which supports flexible natural-language control, and the majority IFEval tasks evaluate this quality of the model.
> >
> > We appreciate the reviewer’s insightful question and want to highlight that this ratio might vary across datasets and domains.  While our preliminary study highlights the sensitivity of this ratio, a thorough investigation across datasets and domains is an exciting direction for future work.

---

> > > ### Author Response · Authors · 2025-11-27
> > > **Response to the rebuttal [Part 3]**
> > >
> > > &nbsp;
> > >
> > > **- What is D_{LLQ}? at Line 171-172? It seems not defined.**
> > >
> > > We are grateful to the reviewer for pointing out this typo. We have updated the manuscript by replacing $\mathcal{D}\_{\mathrm{LLQ}}$ with $\mathcal{D}\_{\mathrm{LDQ}}$ (Line 171-172).
> > >
> > > &nbsp;
> > >
> > > ---
> > > **- Line 349-350, could you explain why D_{SHQ} is mentioned, should it be D_{LMQ}?**
> > >
> > > We appreciate the check and confirm that the reference to $\mathcal{D}\_{\mathrm{SHQ}}$ is correct. In this section, we intentionally use $\mathcal{D}\_{\mathrm{SHQ}}$ because our goal is to compare the effect of early reasoning injection under the same SFT and RL setup.
> > >
> > > We take the no-reason base model $\mathcal{M}\_{\mathrm{base}}$​, fine-tune it on the best-performing SFT dataset $\mathcal{D}\_{\mathrm{SHQ}}$​, and apply RL using Nemotron-CrossThink data. We then compare it with the reasoning-based model $\mathcal{M}\_{\mathrm{LMQ}}$​, which is fine-tuned on the same $\mathcal{D}\_{\mathrm{SHQ}}$​ and trained with the same RL recipe. This ensures a fair comparison where the only difference between the two models is their pretraining data distribution. We have further refined our draft to highlight this explicitly in Line 351-352.

---

### Official Review · Reviewer_6sC9 · 2025-11-01

**Soundness:** 3
**Presentation:** 3
**Contribution:** 3
**Rating:** 6
**Confidence:** 4

**Summary:**

The paper studies when to introduce reasoning-style data in the LLM pipeline. Using an 8B hybrid Mamba-Transformer trained from scratch for 1T tokens, the authors compare injecting reasoning data during pretraining vs. during SFT (and then RL). Core findings: (1) adding diverse reasoning data early (during pretraining) creates durable advantages that post-training cannot “catch up” to (reported ~19% average gain on expert benchmarks); (2) an asymmetric allocation principle: diversity & scale matter most in pretraining, while quality (long CoT) dominates SFT; (3) naïvely scaling mixed-quality SFT data can hurt reasoning; (4) some high-quality pretraining effects are “latent” and only unlock after SFT.

**Strengths:**

Comprehensive empirical study. Thorough, well-controlled experiments quantify how reasoning data affects both pretraining (PT) and post-training (SFT/RL).

Actionable principle. Clear rule of thumb: prioritize diversity and scale of reasoning signals during PT; prioritize quality (e.g., long, high-fidelity CoT) during SFT—immediately useful for practitioners.

Cross-stage validation. Systematic swaps across PT/SFT/RL (controlling data type/quality/scale) repeatedly confirm the “front-loading” thesis, with consistent gains (Tables 1–6).

Informative negative result. Simply doubling mixed-quality SFT data provides no benefit and can degrade math performance, underscoring that quantity cannot replace quality.

Latent PT benefit. A small amount of high-quality reasoning introduced in PT yields improvements that only manifest after SFT, clarifying stage interactions.

**Weaknesses:**

External validity & dataset specificity. The pretraining relies on NVIDIA-curated corpora (~6.2T tokens) and SFT sets such as OpenThoughts and Nemotron-Pretraining-SFT-v1. It’s unclear how well the conclusions transfer to other pretraining bases or fully open corpora.

Clarity on PT/SFT data mixing and table notation. For Tables 2–3, please spell out the exact SFT composition used during post-training for each pretraining variant and standardize the notation. In Table 4, the meaning of SFT×2 is ambiguous: is the dataset literally duplicated (two passes over identical examples) or expanded with additional unique samples? If duplicated, did the model simply see the same data twice (risking overfitting), and was the token budget controlled elsewhere?

Breadth vs. alignment trade-off. Length-filtered long-CoT SFT (DALF) appears to improve reasoning while weakening instruction-following. Any concrete hypothesis (e.g., length effects, stylistic drift, or distribution shift) plus diagnostics on this?


Pretraining mixture definition (80% D_base / 20% D_res). The paper states: “we pretrain from scratch for 1T tokens using 80% D_base and 20% D_res.”  how is that 20% constructed on different reasoning dataset settings?  If any sampling strategy is used? Please clarify this.

**Questions:**

see weakness

---

> ### Author Response · Authors · 2025-11-25
> **Thank you for the encouraging review and the detailed feedback!**
>
> We thank the reviewer for their positive assessment and constructive feedback. We are encouraged that you found our empirical study comprehensive and actionable for practitioners. We also appreciate your recognition of our cross-stage validation experiments and the significance of the negative results regarding naive SFT scaling. We believe all your questions can be addressed within this discussion period, but we would love to provide further clarification if needed and will incorporate the feedback in our final manuscript.
>
> &nbsp;
>
> ---
>
> **- External validity & dataset specificity. It’s unclear how well the conclusions transfer to other pretraining bases or fully open corpora.**
>
> Thank you for raising this concern. To clarify, all data used in our work including the pretraining [[Link](https://huggingface.co/collections/nvidia/nemotron-pre-training-dataset)] and post-training data [SFT: [link](https://huggingface.co/datasets/nvidia/Nemotron-Pretraining-SFT-v1), RL: [link](https://huggingface.co/datasets/nvidia/Nemotron-CrossThink)] come from fully open corpora to ensure reproducibility. To further elaborate on the pretraining corpora, the corpora already contains commonly used open corpora like Common Crawl [1], Arxiv, Wikipedia, StackExchange, Github, OpenWebText [2], OpenWebMath[3] etc.
>
> To provide a holistic understanding of our approach across model sizes and architectures, we further extend our experiments to a transformers model $\mathcal{M}$ of 1.2B parameters for 125B tokens and observe the effect of early reasoning exposure. While pretraining $\mathcal{M}\_{\mathrm{LMQ}}$, we maintain the same token ratio between $\mathcal{D}\_{\mathrm{base}}$ and $\mathcal{D}\_{\mathrm{LMQ}}$ as the previous training setup.
>
> &nbsp;
>
> | **Model**                     | MATH$_{\mathrm{PT}}$ AVG | SCIENCE$_{\mathrm{PT}}$ AVG | CODE$_{\mathrm{PT}}$ AVG | GPR$_{\mathrm{PT}}$ AVG | **Overall** |
> |------------------------------|:------------:|:---------------:|:------------:|:---------------:|:---------------:|
> | $\mathcal{M}_{\mathrm{base}}$ |     6.92     |      18.91      |     12.82     |      40.92      |      19.89      |
> | $\mathcal{M}_{\mathrm{LMQ}}$  |    **16.39** |    **21.50**    |   **23.44**   |    **41.69**    |    **25.75**    |
>
> &nbsp;
>
> As shown in the table above, even with a transformer model (different architecture), the findings hold for a 1.2B model (different scale) trained for a much smaller token horizon (different training token budget). The gains of models trained with reasoning data is significantly higher specifically in reasoning tasks such as math (+9.47%), science (+2.60%) and code (+10.62%) tasks. Even inclusion of reasoning data does not perturb the performance of non-reasoning benchmarks (GPR$_{\mathrm{PT}}$ AVG). These results serve as strong empirical evidence that the proposed data strategy is effective across varying model scales and the widely adopted Transformer architecture. We are grateful to the reviewer for this insightful suggestion. We will include this result in our paper to make our findings stronger and generalizable.
>
> &nbsp;
>
> [1] https://commoncrawl.org/
>
> [2] https://skylion007.github.io/OpenWebTextCorpus/
>
> [3] https://huggingface.co/datasets/open-web-math/open-web-math

---

> ### Author Response · Authors · 2025-11-25
> **Response to the rebuttal [Part 2]**
>
> &nbsp;
>
> **- Clarity on PT/SFT data mixing and table notation. For Tables 2–3, please spell out the exact SFT composition used during post-training for each pretraining variant and standardize the notation.**
>
>
> We appreciate the comment about notation clarification. To reiterate the data mixing across training phases:
>
> + **Pretraining:** We pretrain the $\mathcal{M}\_{\mathrm{base}}$ model solely on the $\mathcal{D}\_{\mathrm{base}}$ data for 1 trillion tokens. To correctly state the data distribution for reasoning models, we pretrain all models for 600B tokens using $\mathcal{D}\_{\mathrm{base}}$ followed by 400B tokens on a mixture of 80% $\mathcal{D}\_{\mathrm{base}}$ and 20% $\mathcal{D}\_{\mathrm{res}}$. This results in a constant budget of 80B reasoning tokens across all experiments. $\mathcal{D}\_{\mathrm{res}}$ can be any one source of data among the three reasoning datasets we have defined in Section 2.2 ($\mathcal{D}\_{\mathrm{SHQ}}$, $\mathcal{D}\_{\mathrm{LDQ}}$, $\mathcal{D}\_{\mathrm{LMQ}}$).
>
> + **SFT:** During SFT, we choose each pretrained model and run SFT on each of the three reasoning datasets ($\mathcal{D}\_{\mathrm{SHQ}}$, $\mathcal{D}\_{\mathrm{LDQ}}$, $\mathcal{D}\_{\mathrm{LMQ}}$) in parallel. Here, we do not mix any data sources for any of the SFT runs.
>
> To summarize, for each pretrained model, we run SFT in parallel across the three reasoning datasets and report the average accuracy in Table 2. We follow the formulation below while presenting the results of Table 2.
>
> + $\mathcal{M}\_{\mathrm{base}}+\mathrm{SFT}$ = Average($\mathcal{M}\_{\mathrm{base}}+\mathrm{SFT}\_{\mathrm{SHQ}}$, $\mathcal{M}\_{\mathrm{base}}+\mathrm{SFT}\_{\mathrm{LDQ}}$, $\mathcal{M}\_{\mathrm{base}}+\mathrm{SFT}\_{\mathrm{LMQ}}$)
>
> + $\mathcal{M}\_{\mathrm{res}}+\mathrm{SFT}$ = Average($\mathcal{M}\_{\mathrm{SHQ}}+\mathrm{SFT}\_{\mathrm{SHQ}}$, $\mathcal{M}\_{\mathrm{SHQ}}+\mathrm{SFT}\_{\mathrm{LDQ}}$, $\mathcal{M}\_{\mathrm{SHQ}}+\mathrm{SFT}\_{\mathrm{LMQ}}$,
> $\mathcal{M}\_{\mathrm{LDQ}}+\mathrm{SFT}\_{\mathrm{SHQ}}$, $\mathcal{M}\_{\mathrm{LDQ}}+\mathrm{SFT}\_{\mathrm{LDQ}}$, $\mathcal{M}\_{\mathrm{LDQ}}+\mathrm{SFT}\_{\mathrm{LMQ}}$,
> $\mathcal{M}\_{\mathrm{LMQ}}+\mathrm{SFT}\_{\mathrm{SHQ}}$, $\mathcal{M}\_{\mathrm{LMQ}}+\mathrm{SFT}\_{\mathrm{LDQ}}$, $\mathcal{M}\_{\mathrm{LMQ}}+\mathrm{SFT}\_{\mathrm{LMQ}}$)
>
>
> We thank the reviewer for pointing this out. We have revised the draft to reflect the result composition (Line 316-317) and the individual results in Table 11. We would like to highlight that each case of $\mathcal{M}\_{\mathrm{res}}$ is better than every case of $\mathcal{M}\_{\mathrm{base}}$. We present the averages to track the overall trend.
>
>
> In Table 3, we use only $\mathcal{M}\_{\mathrm{base}}$ and $\mathcal{M}\_{\mathrm{LMQ}}$ pretrained models, apply SFT with the best performing dataset ($\mathcal{D}\_{\mathrm{SHQ}}$), and then run RL on the resulting SFT model. We highly appreciate the note regarding notation inconsistencies and have standardized them in the revised draft (Line 343-344).
>
>
> &nbsp;
>
> ---
> **- In Table 4, the meaning of SFT×2 is ambiguous: is the dataset literally duplicated (two passes over identical examples) or expanded with additional unique samples?**
>
> To clarify, we run SFT for 2x more epochs than the earlier setup using the same data which means that we see the same SFT data twice compared to the prior setup. We thank the reviewer for pointing out the ambiguity, we have refined this in **Line 373-374** of the revised draft.

---

> ### Author Response · Authors · 2025-11-25
> **Response to the rebuttal [Part 3]**
>
> **- If duplicated, did the model simply see the same data twice (risking overfitting), and was the token budget controlled elsewhere?**
>
> In addition to the SFT for 2x more epochs experiment (which can raise the question of overfitting), we examined whether the no-reason base could catch up to the reason base by giving it more SFT compute using unique tokens instead of repeated ones. To form a baseline, we start from the no-reason base $\mathcal{M}\_{\mathrm{base}}$ and fine-tune it on both $\mathcal{D}\_{\mathrm{LDQ}}$ and $\mathcal{D}\_{\mathrm{ALF}}$, totaling 268M unique samples. We then compare this against the reason base $\mathcal{M}\_{\mathrm{LDQ}}$ which is fine-tuned only on $\mathcal{D}\_{\mathrm{ALF}}$ (7.1M unique samples). Both models are evaluated under the same SFT evaluation setup used in the main paper.
>
> &nbsp;
>
> | Model                              | MATH$_{\mathrm{SFT}}$ AVG | SCIENCE$_{\mathrm{SFT}}$ AVG | CODE$_{\mathrm{SFT}}$ AVG | INS$_{\mathrm{SFT}}$ AVG | Overall |
> |------------------------------------|:------:|:---------:|:------:|:-------------:|:---------:|
> | $\mathcal{M}\_{\mathrm{base}}+ \mathrm{SFT}\_{\mathrm{LDQ+ALF}}$    | 33.66   | 29.15      | 3.49    | 56.86          | 30.79      |
> | $\mathcal{M}\_{\mathrm{LDQ}}+ \mathrm{SFT}\_{\mathrm{ALF}}$      | 60.95   | 47.29      | 22.54   | 39.87          | 42.66      |
>
> &nbsp;
>
> Despite the large gap in unique SFT samples, the reason base achieves a 39% relative gain in Overall and consistently outperforms the no-reason base across all domains. This highlights that strong reasoning foundations built during pretraining cannot be recovered through additional SFT, even when the no-reason base receives considerably more unique data.
>
> &nbsp;
>
> ---
>
> **- Breadth vs. alignment trade-off. Length-filtered long-CoT SFT ($\mathrm{D}\_{\mathrm{ALF}}$) appears to improve reasoning while weakening instruction-following.**
>
> We have noticed this trend across the models  (Table 5)---less diverse data weakens the instruction following ability. We hypothesize that this degradation is driven by distribution shift and stylistic rigidity as $\mathrm{D}\_{\mathrm{ALF}}$ is heavily skewed toward Math and Code, similar to $\mathrm{D}\_{\mathrm{SHQ}}$.  To further investigate, we compared the instruction-wise accuracy of  $\mathcal{M}\_{\mathrm{LDQ}}+\mathrm{SFT}\_\mathrm{ALF}$ (less diverse) and $\mathcal{M}\_{\mathrm{LDQ}}+\mathrm{SFT}\_\mathrm{LDQ}$ (diverse) on IFEval benchmark [1].
>
> &nbsp;
>
> | Instruction Type | $\mathcal{M}\_{\mathrm{LDQ}}+\mathrm{SFT}\_\mathrm{ALF}$ | $\mathcal{M}\_{\mathrm{LDQ}}+\mathrm{SFT}\_\mathrm{LDQ}$ | Diff |
> |----------------|--------------|---------|------|
> | punctuation:no_comma | 13 | 27 | 14 |
> | length_constraints:number_words | 19 | 30 | 11 |
> | change_case:english_lowercase | 19 | 28 | 9 |
> | keywords:letter_frequency | 13 | 21 | 8 |
> | change_case:english_capital | 9 | 16 | 7 |
> | language:response_language | 19 | 26 | 7 |
> | detectable_format:number_bullet_lists | 14 | 21 | 7 |
> | combination:two_responses | 11 | 16 | 5 |
> | keywords:forbidden_words | 12 | 17 | 5 |
> | detectable_format:title | 32 | 36 | 4 |
> | startend:quotation | 19 | 23 | 4 |
> | change_case:capital_word_frequency | 12 | 15 | 3 |
> | length_constraints:number_paragraphs | 9 | 12 | 3 |
> | length_constraints:nth_paragraph_first_word | 3 | 5 | 2 |
> | length_constraints:number_sentences | 29 | 31 | 2 |
> | detectable_format:json_format | 13 | 15 | 2 |
> | startend:end_checker | 11 | 13 | 2 |
> | detectable_format:number_highlighted_sections | 40 | 41 | 1 |
> | detectable_format:constrained_response | 9 | 10 | 1 |
> | detectable_content:number_placeholders | 23 | 23 | 0 |
> | combination:repeat_prompt | 17 | 17 | 0 |
> | detectable_format:multiple_sections | 12 | 12 | 0 |
> | detectable_content:postscript | 23 | 22 | -1 |
> | keywords:existence | 31 | 30 | -1 |
> | keywords:frequency | 29 | 26 | -3 |
>
> &nbsp;
>
> From the table, we can see that $\mathrm{D}\_{\mathrm{LDQ}}$ significantly improves performance of Linguistic Manipulation (Punctuation & Case) instructions. This is likely because diverse data exposes the model to a wider range of writing styles, formats, and Markdown structures, which supports flexible natural-language control. In contrast, math and code-heavy data, $\mathrm{D}\_{\mathrm{ALF}}$  reinforce strict token precision, helping with tasks involving keyword presence, token counting, or rigid pattern detection.
>
> In short, diverse data boosts adaptability and linguistic control, helps the model understand how to say things (style, format, length, punctuation), and allows it to break standard rules when asked. While less diverse data enforces precision. It is excellent for "hard" constraints like ensuring a specific word exists or counting specific items, but it lacks the stylistic range to handle "soft" linguistic constraints. We thank the reviewer for the insightful question and will share these insights in our final draft.
>
> [1] https://huggingface.co/datasets/google/IFEval

---

> > ### Author Response · Authors · 2025-11-25
> > **Response to the rebuttal [Part 4]**
> >
> > **- Pretraining mixture definition. How is that 20% constructed on different reasoning dataset settings? If any sampling strategy is used?**
> >
> > While dealing with each reasoning datasets pointing towards each axis (1. Small-Scale, High-Quality Data, $\mathcal{D}\_{\mathrm{SHQ}}$, 2. Large-Scale, Mixed-Quality Data, $\mathcal{D}\_{\mathrm{LMQ}}$ and 3. Large-Scale, Diverse Data, $\mathcal{D}\_{\mathrm{LDQ}}$), we only pretrain/run SFT with one type of data. To be specific, we do not combine these datasets while preparing 20% reasoning data for training, rather we only pick one data for each training to pinpoint the effect of data with unique features. For example,
> >
> > * $\mathcal{M}\_{\mathrm{SHQ}}$ has been trained with $\mathcal{D}\_{\mathrm{base}}$ and $\mathcal{D}\_{\mathrm{SHQ}}$
> >
> > * $\mathcal{M}\_{\mathrm{LDQ}}$ has been trained with $\mathcal{D}\_{\mathrm{base}}$ and $\mathcal{D}\_{\mathrm{LDQ}}$
> >
> > * $\mathcal{M}\_{\mathrm{LMQ}}$ has been trained with $\mathcal{D}\_{\mathrm{base}}$ and $\mathcal{D}\_{\mathrm{LMQ}}$
> >
> > We thank the reviewer for the clarification question. We have revised Section 2.2 and Appendix A to reflect these details accurately and explicitly.

---

### Author Response · Authors · 2025-11-29
**Summary of the feedback and revisions**

We thank the AC, PC, and SPCs for their time and effort. We are encouraged by the positive consensus among all four reviewers, noting that their inquiries were limited to clarification questions that were easily addressed in the rebuttal. We have already updated the manuscript with all the additional clarifications. We summarize the major feedback and our key revisions below:


**Summary of Positive Feedback**

* **Comprehensive and systematic empirical study (Reviewers 6sC9, 2v5m, WLmB, hBYq):** The experiments are praised for being thorough, well-controlled, and conducted at a large scale across diverse downstream tasks and data regimes.
* **Actionable training principles (Reviewers 6sC9, hBYq):** The finding of an "asymmetric allocation principle"—prioritizing diversity and scale in pretraining while prioritizing quality/depth in SFT—is considered a clear, practical, and immediately useful guideline for practitioners.
* **Novel and impactful investigation (Reviewers 6sC9, WLmB, hBYq):** The study challenges the conventional practice of adding reasoning data only during post-training, providing a novel and timely perspective on "front-loading" these capabilities.
* **Insightful analysis of stage interactions (Reviewers 6sC9, 2v5m, WLmB, hBYq):** Reviewers appreciated the rigorous ablation studies, particularly the identification of "latent" pretraining benefits that only unlock after SFT, as well as the informative negative results regarding naïve SFT scaling.

**Summary of the Key Revisions**

* **Validation across Architecture and Scale (Reviewers 6sC9, WLmB, hBYq):** We extended our experiments to a 1.2B parameter Transformer model trained for 125B tokens to test generalizability beyond the hybrid architecture and model scale.
    * *Finding:* The benefits of early reasoning exposure hold true for the Transformer architecture, showing significant 29.46% relative gains on average across diverse downstream tasks.
* **Impact of Data Ratio (Reviewer 2v5m):** We conducted an ablation study testing different ratios (80/20, 90/10, 60/40 base vs reasoning tokens)
    * *Finding:* Performance scales positively with the volume of reasoning data; increasing the ratio to 40% yielded the highest results, while reducing it to 10% caused a mild decline.
* **Diversity vs. Quality in Pretraining (Reviewer 2v5m):** We performed a controlled experiment using a downsampled dataset ($\mathcal{D}_{\mathrm{ALF}}$) that retained only high-quality, long-context examples but lacked diversity and scale.
    * *Finding:* Reducing diversity and scale resulted in a significant performance drop (~11.5 points), validating that pretraining relies heavily on breadth (diversity/scale) rather than just sample complexity.
* **SFT Catchup with more Unique tokens (Reviewer 6sC9):** We compared our reason-base model against a no-reason base model that received significantly more unique tokens during the SFT stage (rather than repeated epochs).
    * *Finding:* The reason-base model consistently outperformed the no-reason base despite the later seeing ~37x more unique SFT samples, proving that pretraining advantages cannot be "caught up" simply by scaling SFT data quantity.

We sincerely thank the reviewers for their insightful feedback, which has helped strengthen our work. We have addressed each comment in detail in the respective sections below.

---

> ### Author Response · Authors · 2025-12-01
> **Summary of the Updates in the Discussion Period**
>
> **Summary of the Updates in the Discussion Period**
>
> * We want to highlight that during the discussion period, **Reviewer WLmB and hBYq further raised their scores** after addressing their concerns.
> * Reviewer 6sC9 and 2v5m both started with positive scores (6, 6). We have thoroughly addressed all the clarifications during the discussion period. Unfortunately, there was no further discussion from them before November 27.

---

### Meta-Review · Area_Chair_tEZh · 2026-01-10

**Summary:**

This paper provides a large, carefully controlled empirical study of when to introduce reasoning data in the LLM training pipeline. Across multiple reasoning data regimes and across stages, the central conclusion is consistent: front-loading reasoning data into pretraining yields durable gains that later SFT cannot fully recover, even when SFT is scaled. Reviewers generally found the experimental design thorough, the ablations informative, and the resulting “asymmetric allocation principle” practically useful: pretraining benefits most from broad diversity and coverage of reasoning patterns, while SFT is more sensitive to high-quality, deep reasoning traces.

During rebuttal/discussion, the authors clarified key experimental details and added new experiments to address external validity and confounding concerns, including (i) controlled downsampling to disentangle diversity/scale effects, (ii) ratio ablations, and (iii) additional validation on a Transformer at a different scale/token budget.

**Reviewer Concerns:**

Major concerns addressed by rebuttal/discussion
1. Clarity of training/data setup
2. Diversity vs quality in pretraining confounding
3. Architectural / scale generalization of results
4. Evaluation on non-reasoning (memory) tasks

Concerns still outstanding
1. Mechanistic explanation is partial
2. Trade-offs with instruction following/alignment

**Reviewer Scores:**

6sC9: 6 → 6 or 7 .

2v5m: 6 → 6 or 7

WLmB: 4 → 5 or 6. Reviewer explicitly stated they would raise their score after concerns were solved.

hBYq: 4 → 5 or 6. Reviewer explicitly stated all concerns were addressed and they would raise their score.

---

### Decision · Program_Chairs · 2026-01-26

Accept (Poster)